# On-demand seizures facilitate rapid screening of therapeutics for epilepsy

Yuzhang Chen[1,2], Brian Litt[3,4,5], Flavia Vitale[2,3,4,5], Hajime Takano[4,6]*

[1]Department of Neuroscience, Perelman School of Medicine, University of Pennsylvania, Philadelphia, United States; [2]Center for Neurotrauma, Neurodegeneration, and Restoration, Corporal Michael J. Crescenz Veterans Affairs Medical Center, Philadelphia, United States; [3]Department of Bioengineering, University of Pennsylvania, Philadelphia, United States; [4]Department of Neurology, Perelman School of Medicine, University of Pennsylvania, Philadelphia, United States; [5]Center for Neuroengineering and Therapeutics, University of Pennsylvania, Philadelphia, United States; [6]Division of Neurology, Department of Pediatrics, The Children's Hospital of Philadelphia, Philadelphia, United States

## eLife Assessment

The authors modified a common method to induce epilepsy in mice to provide an improved approach to screen new drugs for epilepsy. This is **important** because of the need to develop new drugs for patients who are refractory to current medications. The authors' method evokes seizures to circumvent a low rate of spontaneous seizures and the approach was validated using two common anti-seizure medications. The strength of evidence was **solid**, making the study invaluable, but there were some limitations to the approach and methods.

*For correspondence:
takanoh@chop.edu

Competing interest: The authors declare that no competing interests exist.

**Abstract** Animal models of epilepsy are critical in drug development and therapeutic testing. However, dominant methods for evaluating epilepsy treatments face a tradeoff between higher throughput and etiological relevance. Screening models are either based on acutely induced seizures in wild-type, naive animals or spontaneous seizures in chronically epileptic animals. Each has its disadvantages – acute convulsant or kindling-induced seizures do not account for the myriad neuropathological changes in the diseased, epileptic brains, and spontaneous behavioral seizures are sparse in chronically epileptic models, making it time-intensive to sufficiently power experiments. In this study, we developed the Opto-IHK (optogenetically induced seizures in intrahippocampal kainate mice) model, a mechanistic approach to precipitate seizures 'on demand' in chronically epileptic mice. We briefly synchronized principal cells in the CA1 region of the diseased hippocampus to reliably induce stereotyped on-demand behavioral seizures. These induced seizures resembled naturally occurring spontaneous seizures in the epileptic animals and could be stopped by commonly prescribed anti-seizure medications such as levetiracetam and diazepam. Furthermore, we showed that seizures induced in chronically epileptic animals differed from those in naive animals, highlighting the importance of evaluating therapeutics in the diseased circuit. Taken together, we envision the Opto-IHK model to accelerate the evaluation of both pharmacological and closed-loop interventions for epilepsy.

## Introduction

Epilepsy, a set of neurological disease syndromes characterized by recurrent, spontaneous seizures, is a debilitating condition that affects millions of people worldwide (*Asadi-Pooya et al., 2023*; *Fiest*

*et al., 2017*). Even with the development of over 40 anti-seizure medications (ASMs) over the course of the last century, between 15% and 30% of patients are unable to achieve seizure freedom (*Kalilani et al., 2018*; *Dalic and Cook, 2016*; *Sultana et al., 2021*). The percentage of patients with drug-resistant seizures has remained constant despite the introduction of multidrug therapies and newer ASMs classes (*Löscher et al., 2020*).

The lack of screening in etiologically relevant models may be one reason for the disconnect between the increasing types of medications and the stubbornness of drug-resistant seizures to treatment (*Klitgaard et al., 1998*; *Wilcox et al., 2020*; *Löscher and White, 2023*). Many patients with drug-resistant seizures have temporal lobe epilepsy (TLE), a type of epilepsy where seizures originate from temporal lobe structures such as the hippocampus (*Asadi-Pooya et al., 2017*). The brains of TLE patients typically have structural and molecular changes, including hippocampal sclerosis, axonal sprouting, and receptor alterations (*Tai et al., 2018*; *Ying et al., 1998*; *Brooks-Kayal et al., 1999*; *Loup et al., 2000*). The resulting epileptic network functions differently from that of the healthy brain. This divergence may be why ASMs that work in a naive, wild-type animal screen are ineffective in epileptic patients.

Fortunately, TLE is well modeled in animals (*Leite et al., 2002*; *Kandratavicius et al., 2014*; *Rusina et al., 2021*; *Lévesque et al., 2021*). The intrahippocampal kainate (IHK) model of epilepsy captures key markers of human TLE – animals exhibit hippocampal sclerosis, mossy fiber sprouting, and spontaneous limbic seizures (*Venceslas and Corinne, 2017*; *Leite et al., 1996*). Despite its lengthy history, the IHK model was only recently introduced into the Epilepsy Therapy Screening Program (ETSP) pipeline (*Wilcox et al., 2020*; *Kehne et al., 2017*). There were many factors contributing to this delay. Spontaneous seizures, by their very nature, are sparse, irregular, and unpredictable (*Puttachary et al., 2015*; *Williams et al., 2009*; *Rattka et al., 2013*). Compared to testing ASMs in wild-type animals by acutely inducing seizures via convulsants or kindling, testing therapeutics in etiologically relevant models is often a much more lengthy and unwieldy process.

To address the challenges of using etiologically relevant models for therapy screening, we asked whether we could harness the epileptic circuit to generate seizures on demand. Of particular interest to seizure initiation and propagation is region CA1 of the hippocampus. The CA1 is the main output of the hippocampus in the canonical tri-synaptic circuit (*Zutshi et al., 2022*). In vitro, electrical activation of excitatory inputs to the CA1 can generate seizure-like bursting (*Meier and Dudek, 1996*). In animals, calcium imaging demonstrates that CA1 principal cells are highly active during acute convulsant-induced seizures (*Mulcahey et al., 2022*). In addition, changes in the CA1 cellular population during epileptogenesis, such as an increase in burster cells, make the CA1 predisposed to rapid firing that can elicit epileptic events (*Chen et al., 2011*; *Esclapez et al., 1999*). Thus, we hypothesized that hyperexcitation of CA1 principal cells would activate etiologically relevant mechanisms and initiate seizures in epileptic animals.

Specifically, we investigated whether selective optogenetic activation of CA1 principal cells could precipitate time-locked seizures in freely moving epileptic animals (*Figure 1*). We refer to this approach as the Opto-IHK model (optogenetically induced seizures in intrahippocampal kainate

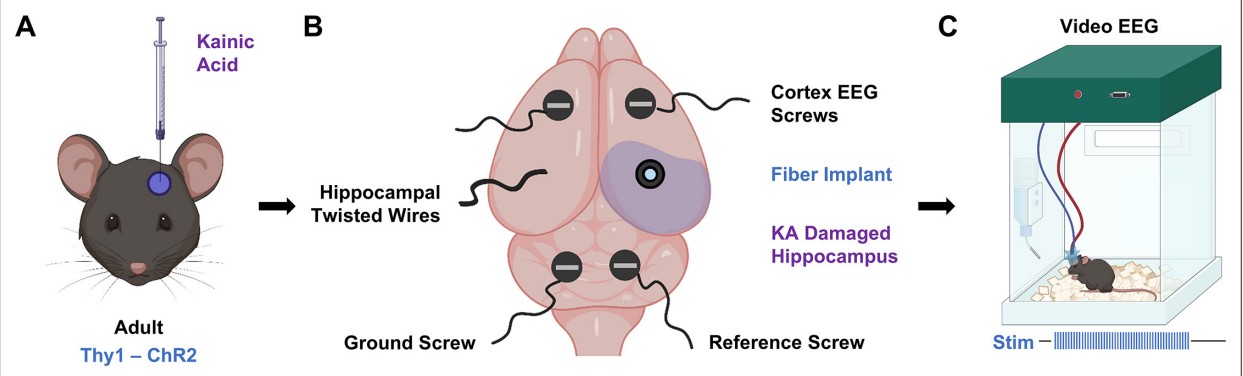

**Figure 1.** Experimental schematic for on-demand seizure induction in epileptic animals. (**A**) Chronic epilepsy was induced in Thy1-ChR2 mice via intrahippocampal kainic acid injection into the CA3. (**B**) Mice were implanted with EEG recording apparatus consisting of two cortical screws, one set of insulated braided wire targeting the hippocampus, a ground screw, and a reference screw. A fiber was positioned so that the tip illuminated the CA1. (**C**) 10 Hz of 473 nm light delivered into the CA1 activated Thy1 ChR2 neurons and induced seizures on demand in vivo.

mice), a mechanistically driven extension of the classic IHK model that preserves CA1 structure and better reflects human TLE pathology. To validate Opto-IHK, we first compared the induced activity to spontaneous seizures. Next, we compared seizures induced in epileptic animals to those in wild-type, naive animals to ascertain whether unique therapeutically relevant features are evident in the epileptic brain. Finally, we attempted to shut down induced seizures with known anti-seizure medications. We present evidence that hypersynchronous excitation of CA1 principal cells in the Opto-IHK model can induce focal to bilateral tonic-clonic seizures in mice, and that these seizures can be used to evaluate the therapeutic efficacy of both pharmacological and time-sensitive treatments.

## Results

To avoid confusion and ensure clarity in the interpretation of the results presented in this manuscript, we define the following terms:

*Afterdischarge*: Afterdischarges, also referred to as induced activity or electrographic events, refer to long-lasting (5 s or more) electrical activity elicited following optical stimulation. This denotation includes both seizures and spikes. We use 'afterdischarges' interchangeably with 'induced activity'.

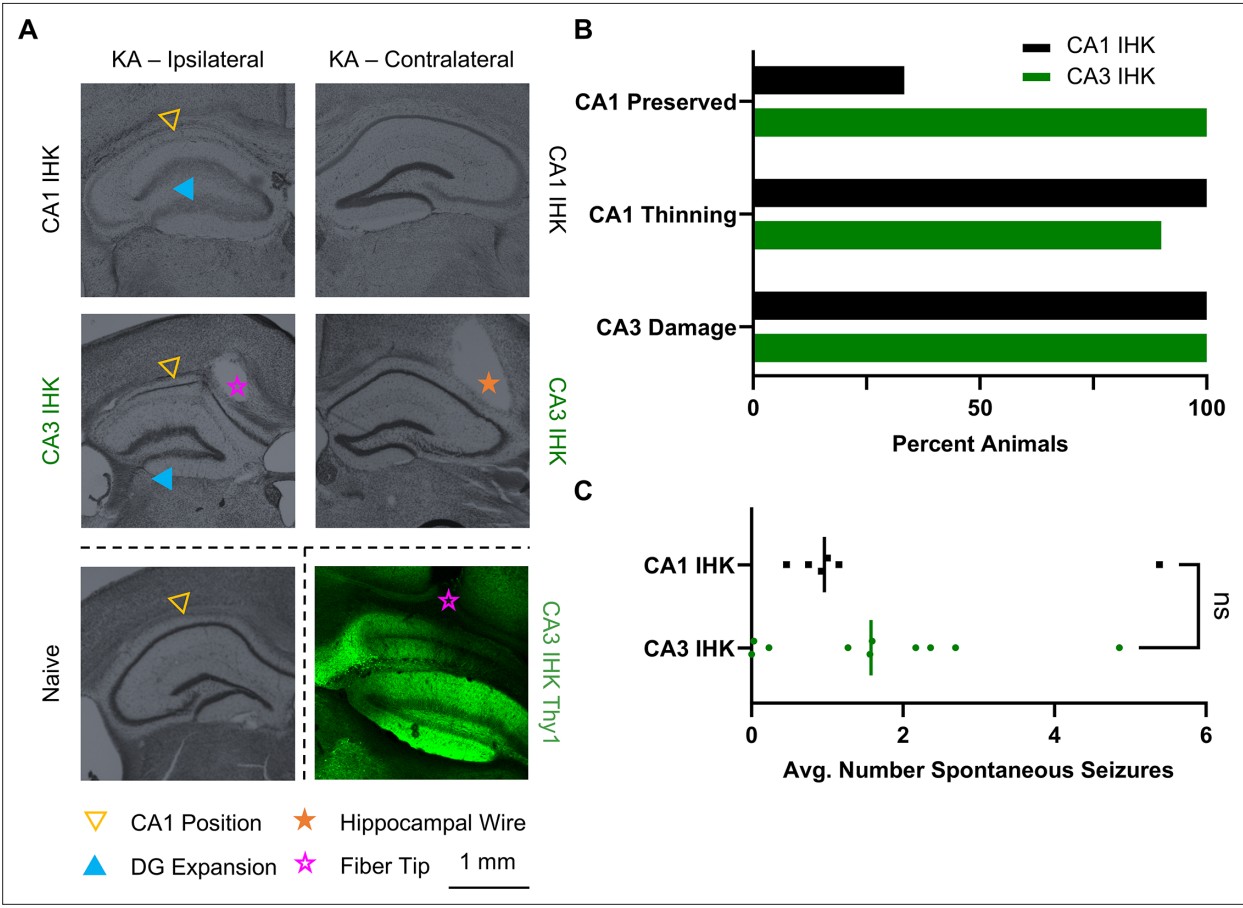

**Figure 2.** Intrahippocampal kainic acid (IHK) injection into the CA3 reduced CA1 damage without impacting average spontaneous seizure count. (**A**) Cresyl violet staining of hippocampal slices from IHK injected animals visualized IHK-induced damage to the hippocampus. The CA1 layer was eliminated in CA1 IHK animals (top). In the CA3 IHK animals, the CA1 layer was present but thinned (middle). Both CA1 IHK and CA3 IHK animals displayed expansion of the dentate gyrus structure and extensive CA3 damage when compared to naive animals (bottom left). Fiber was implanted above cells expressing Thy1-ChR2 in the CA1 (bottom right). (**B**) Two thirds (***Dalic and Cook, 2016***) of CA1-IHK animals did not have a visible CA1 cell layer, while the rest (***Fiest et al., 2017***) had a thinned cell layer. In contrast, all CA3 IHK (***Asadi-Pooya et al., 2017***) animals had a visible CA1 cell layer. All 10 CA3 IHK and 6 CA1 IHK animals had extensive damage to the CA3 coupled with expansion of the dentate gyrus (DG). (**C**) Two-tailed Mann-Whitney test comparing the average number of spontaneous seizures per day showed no significant difference (p=0.635) between CA1 IHK and CA3 IHK animals.

*Seizures*: We use the term 'seizure' specifically to describe generalized seizures that had behavioral components verified on video. Mostly, these are tonic-clonic seizures with Racine Scale 3 or higher, but occasionally we observed milder seizures with Racine Scale 1 or 2.

## CA3 kainate injection preserves CA1 cell layer targeted during on-demand seizure induction

The on-demand seizure induction procedure in mice expressing excitatory channelrhodopsins in CA1 pyramidal neurons (Thy1-ChR2 mice) is a three-part process (*Figure 1*). First, we induced chronic epilepsy in the mice via CA3 IHK injection (*Figure 1A*). An electroencephalogram (EEG) implantation surgery followed, during which two recording screws were inserted into the cortex, and two additional screws – a ground and a reference – were inserted into the cerebellum (*Figure 1B*). An implanted optical fiber targeting the CA1 ipsilateral to the IHK injection allowed for optical excitation at the seizure focus, while a braided wire targeting the CA1 contralateral to IHK injection provided information on inter-hippocampal activity (*Figures 1A and 2B* middle, 2 A bottom right). After the mice recovered for a week, they underwent 24 hr continuous video/EEG monitoring, during which 10 Hz, 473 nm optical stimulation was applied to induce seizures on demand (*Figure 1C*).

The canonical injection site for IHK models is the dorsal CA1 (*Paschen et al., 2020*); however, the destruction of the CA1 cell layer due to IHK injection is fundamentally incompatible with our approach for inducing seizures. Kainate, a glutamate agonist, induces hyperexcitation of cells in the vicinity of the injection site, leading to cell death and gliosis (*Zhang and Zhu, 2011*). The damage is clearly visible in the CA1, and the sclerosis is hypothesized to be the center of seizure generation (*Krook-Magnuson et al., 2015*; *Figure 2A* top). Thy1 expression in the CA1 localizes to principal cells (*Dobbins et al., 2018*), so it is critical to preserve the health of the CA1 cell layer. To do so, we moved the IHK injection site to the medioventral CA3, which is 0.7 mm posterior, 1.4 mm lateral, and 1.6 mm ventral to the canonical CA1 injection site. CA3 IHK animals have a sparse CA1 layer (*Figure 2A* middle) compared to naive animals (*Figure 2A* bottom left), while CA1 IHK animals often do not have a visible CA1 layer (*Figure 2A* top).

Comparative analysis of brain slices extracted from the CA1 IHK (n=6) and the CA3 IHK (n=10) mice showed greater CA1 preservation in the CA3 IHK group (*Figure 2B*). Nissl staining using cresyl violet highlighted neural structures and stained the CA1 principal cell layer. Slices extracted at AP – 2.0 mm from bregma showed that the CA1 cell layer was present in only one-third of CA1 IHK animals. In contrast, the CA1 cell layer was present in all CA3 IHK animals (*Figure 2B* top). In both groups, the CA1 layer tended to be thinned when present. However, the CA1 in 1 CA3 IHK animal did not appear to be thinned (*Figure 2B* middle). All animals in the CA3 IHK and the CA1 IHK groups exhibited CA3 damage and morphological changes in the dentate gyrus (*Figure 2A and B* bottom). Thus, CA3 damage was not unique to the CA3 IHK group; it was already present in animals with the canonical CA1 IHK injection.

While preserving the CA1 structure was important to our model, we wanted to ensure that shifting the IHK injection site to the CA3 did not alter the number of spontaneous seizures an animal experienced per day. Thus, we acquired continuous video-EEG recording on the same CA1 IHK and CA3 IHK animals and tracked spontaneous seizures. We found that the average number of spontaneous seizures a day was similar between the two groups. The CA1 IHK animals experienced an average of 1.6 spontaneous seizures a day, while the CA3 IHK animals experienced an average of 1.7 spontaneous seizures a day. The nonparametric two-tailed Mann-Whitney test, which compared the difference between the median daily spontaneous seizure count of the CA1 IHK (1.0) and CA3 IHK (1.6) animals, also failed to find a significant difference between the two groups (p=0.635; *Figure 2C*).

## Optogenetically induced seizures resemble spontaneous tonic-clonic seizures in epileptic mice

In epileptic, freely moving Thy1-ChR2 mice (n=10, 7 males, 3 females), we observed spontaneous generalized seizures, as exemplified in *Figure 3A*. Over 1 week of baseline recording, animals experienced on average 1.0±1.8 (mean ± standard deviation) behavioral spontaneous seizures per day. We then optically stimulated Thy1-ChR2 expressing neurons in the CA1 ipsilateral to the IHK injection site using 10 Hz, 473 nm light several times a day at one-hour intervals. The threshold laser power and stimulation duration, defined as the minimum necessary for consistent (>66%) induction

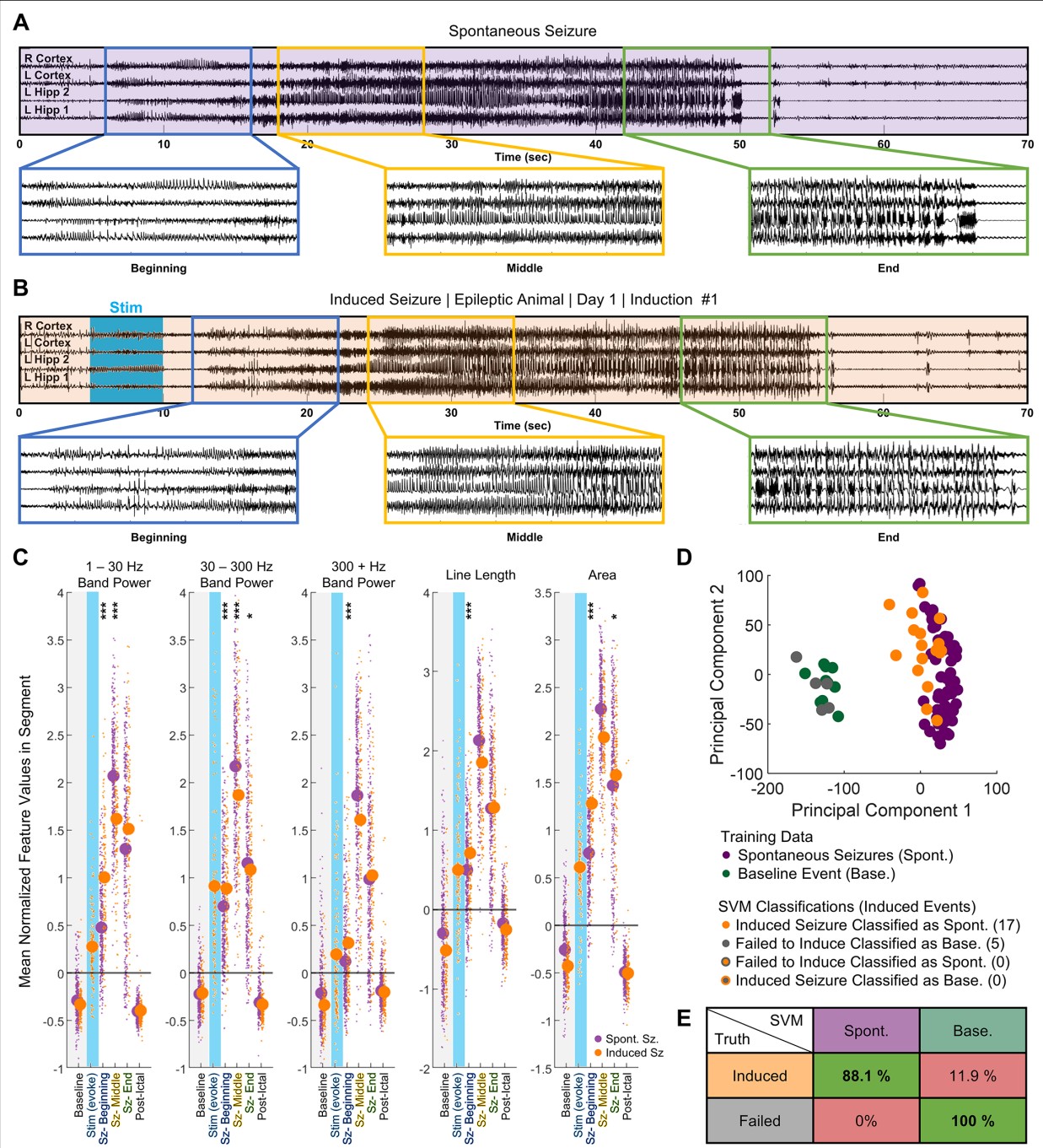

**Figure 3.** Induced seizures resembled naturally occurring spontaneous seizures in chronically epileptic animals. (**A**) Example of electrographic signal from spontaneous seizure in epileptic animal, with three segments of 10 s enlarged for clarity. (**B**) Electrographic signal from first induction in the same animal. (**C**) Change in features from baseline (gray) to computationally defined segments (first tercile – beginning, second tercile – middle, and final tercile – end). In the first tercile, the extent of increase in 1–30 Hz band power, 300+ Hz band power, line length, and area significantly differed between 138 induced (n=10) and 337 spontaneous behavioral seizures (n=8). Differences become less significant by the middle and the final third. Linear Mixed Effect Model: * p<0.05, ** p<0.01, *** p<0.001 (**D**) Linear Support Vector Machine (SVM) classified inductions from the animal in (**A**) and (**B**). Successfully induced seizures were more closely associated with spontaneous seizures than with either baseline activity or optogenetic activations that failed to induce afterdischarges exceeding 5 s. (**E**) Compiled table of SVM accuracies across epileptic animals (n=8). Induced activity was classified as similar to spontaneous seizures in 88.1% of all successful activations. Failed activations were classified as baseline 100% of the time.

The online version of this article includes the following figure supplement(s) for figure 3:

**Figure supplement 1.** Threshold determination for inducing consistent afterdischarges.

**Figure supplement 2.** Examples of behavioral and electrographic seizures induced in epileptic animals.

of afterdischarges lasting a minimum of 5 s, varied between the animals, with an average threshold laser power of 9.14±4.75 mW and an average stimulation duration of 6.30±1.64 s (*Figure 3—figure supplement 1*). Using the threshold stimulus, activity was induced approximately 88.7 ± 8.8% of the time in the epileptic animals. Average duration of induced activity, mainly generalized seizures with behavioral components, was 30.98±4.69 s (*Figure 3—figure supplement 1*). After optical stimulation began, animals experienced 1.8±3.5 spontaneous behavioral seizures per day on average. A two-tailed Wilcoxon rank sum test on the distribution of daily spontaneous seizure counts before and after stimulation did not find that the difference of the two groups' medians diverged from 0 (p=0.1336).

We next sought to determine whether EEG signals differed between spontaneous seizures and induced seizures. First, we analyzed standard EEG features (*Stancin et al., 2021*) such as band power at three different frequency ranges, line length, and area, in overlapping 500 ms sections. These features were normalized to a short pre-stimulation baseline period, after which a K nearest neighbor classifier was employed to determine the seizure durations. Subsequently, we divided the activity into three segments (beginning, middle, ending). This approach allows us to compare features across seizures of varying durations. Our analysis also shows that features vary widely within each animal; to account for the high intra-animal variation, we used an interactive linear mixed effect model that accounted for intra-animal residuals to identify common inter-animal trends over time. This method-ology allows us to identify changes across animals, changing the independent unit in the analysis to animals instead of seizures.

Spontaneous behavioral seizures underwent stereotyped changes throughout the three segments (*Figure 3A*). Visually, we noted that the beginning third of the seizure was characterized by an increase in spiking activity. The middle third was characterized by higher frequency activity and behaviorally resembled the tonic phase of a tonic-clonic seizure (*D'Ambrosio and Miller, 2010*). The final third was characterized by bursts and behaviorally resembled the clonic phase of a tonic-clonic seizure. Next, we quantified the feature space changes within the 337 spontaneous seizures from eight animals using the linear mixed effect model (*Figure 3C*). In the beginning third, there was a significant increase in the area ($Pp<1e^{-4}$), all band power frequencies ($p<1e^{-4}$), and line length ($p<1e^{-4}$) over baseline values. A further increase occurred in all three features ($p<1e^{-4}$) between the beginning and middle third of the seizure before decreasing between the middle and final third of the seizure ($p<1e^{-4}$). The elevated values in the final third were still significantly higher than baseline ($p<1e^{-4}$).

Induced activity predominantly consisted of generalized seizures (*Figure 3B*). Like in the sponta-neous seizures, most of induced seizures underwent three visually distinct phases. In the beginning, there was a general increase in spiking amplitude and the emergence of higher frequency activity that was occasionally accompanied by behaviors such as freezing. The middle portion resembled the tonic phase of a seizure – the animal displayed forelimb clonus, shaking, or stiffening of the tail. In the final third, bursts began to emerge on the EEG signal, and the animal entered the clonic phase of the seizure, during which it uncontrollably reared, backpedaled, jumped, or fell on its side (*Figure 3—figure supplement 2*). Occasionally, optogenetic stimulus induced seizures of lower severity, afterdis-charges without seizure behavior, or failed to cause afterdischarges (*Figure 3—figure supplement 2*). In the feature space, induced seizures followed a similar trend as spontaneous seizures (*Figure 3C*). Line length, area, and band power significantly increased between baseline and the beginning third ($p<1e^{-4}$). A further increase occurred between the beginning and the middle third ($p<1e^{-4}$). Between the middle and final third, a significant decrease in area ($p<1e^{-4}$), line length ($p<1e^{-4}$), and 30 Hz +band power ($p<1e^{-4}$) occurred. However, unlike in spontaneous seizures, the decrease in 1–30 Hz band power between the middle and final third in induced seizures was not significant (p=0.10).

We then compared the progression of spontaneous and induced seizures in the feature space. All confirmed spontaneous seizures had a Racine score of 3 or greater. To ensure a fair comparison, we used two criteria to filter the induced activity prior to comparison: (1) afterdischarges must have lasted a minimum of fifteen seconds and (2) Racine seizure score had to be greater than or equal to 3. 138 induced seizures from 10 animals fit these criteria. As shown in *Figure 3C*, we used a linear mixed effect model to compare the changes in the band power, line length, and area of each segment to the baseline values. In the beginning third, the extent of increase across many features significantly differed between the induced and the spontaneous seizures. Specifically, the increase in area was significantly higher for induced seizures ($p<1e^{-4}$), as were the increases in line length ($p<1e^{-4}$), 1–30 Hz band power ($p<1e^{-4}$), 30–300 Hz band power ($p<1e^{-4}$), and 300–1000 Hz band power ($p<1e^{-4}$). By

the middle third, significant differences remained in the 1–30 Hz band power (p<1e⁻⁴) and 30–300 Hz band power (p=0.0001). However, the relationship was now inverted – the 1–300 Hz band power increased more for spontaneous seizures than for induced seizures. The changes in other features, such as 300–1000 Hz band power (p=0.3866), line length (p=0.7649), and area (p=0.1962), were no longer statistically significant. In the final third, the increase in area (p=0.0259) and 30–300 Hz band power (p=0.0160) were slightly significant. The increase in 1–30 Hz band power was no longer significantly different (p=0.073), and the increases in 300–1000 Hz band power (p=0.3245) and line length (p=0.1172) remained not significant.

To further quantify the similarity between spontaneous and induced activity, we trained an individual one-class support vector machine (SVM) classifier to classify whether the induced activity was a seizure or not. The SVM is trained exclusively on data from spontaneous seizures and baseline activity; as such, the SVM is not designed to differentiate between induced and spontaneous seizures. Induced activity is then presented to the SVM, which classifies it as either in the category of spontaneous seizures or baseline activity. A sample classification output for one animal is shown in *Figure 3D*. In this animal, all optogenetic activations that resulted in a minimum afterdischarge length of 5 s were classified as spontaneous seizures by the classifier. Meanwhile, inductions that failed to induce afterdischarges lasting more than 5 s were all classified as baseline. When we integrated the results from all 8 SVM classifiers, we found that activations that induced afterdischarges lasting longer than 5 s were classified as spontaneous seizures 88.1% of the time, whereas 11.9% of the time they were inaccurately classified as baseline. Meanwhile, activations that failed to induce afterdischarges longer than 5 s were classified as baseline 100% of the time (*Figure 3E*). Taken together, the results show that synchronizing CA1 principal cell activity in epileptic mice generated seizures that resembled naturally occurring spontaneous seizures, both behaviorally and on EEG, with the minor exception of EEG features during seizure onset.

## Induced activity significantly differs between epileptic and naive animals

During epileptogenesis, neural networks in the brain undergo various changes ranging from the modification of membrane receptors to the formation of new synapses (*Vergaelen et al., 2024*; *Goldberg and Coulter, 2013*; *Godale and Danzer, 2018*). We hypothesized that these changes are critical for successful on-demand seizure induction. To test our hypothesis, we attempted to optically induce seizures in naive, wild-type Thy1-ChR2 mice and compared the induced activity in naive mice to the induced activity in epileptic mice.

Initial optical provocations in naive animals (n=7, 3 males, 4 females) typically caused low frequency, high amplitude spiking activity (*Figure 4A*). While long-lasting, these afterdischarges did not have a behavioral seizure manifestation. Animals displayed independent movement, exploration, or grooming during the afterdischarges (*Figure 4—figure supplement 1*). Behavioral differences between initial naive and epileptic inductions did not result from changes in the optical induction parameters – the naive threshold laser power (6.17±1.58 mW) and stimulation duration (5.67±1.03 s) both did not significantly differ from that of the epileptic animals (p>0.05; *Figure 3—figure supplement 1*). Rather, the observed behavior in the naive animals was likely due to a lack of etiologically relevant changes in the healthy brain.

After 4 consecutive days of up to 5 optogenetic activations per day at a frequency of 1 per hour, we noticed that applying the same optogenetic stimulus resulted in visibly more higher frequency activity and bursting (*Figure 4B*). In addition, all seven animals displayed stereotypical behavioral signs of seizure, including shaking, tail stiffening, rearing, wild running, and uncontrolled jumping (*Figure 4—figure supplement 1*). The increased likelihood of behavioral seizures in later inductions is reminiscent of the 'kindling model' of epilepsy, where repeated administration of subthreshold electrical stimulation eventually causes the animal to enter a 'kindled' state. In the kindled state, the previously subthreshold electrical stimulation can induce seizures (*Chen et al., 2016*).

To visualize the 'kindling effect' of the optogenetic stimulation, we evaluated the rate at which both electrographic activity and seizure-like behaviors were induced over 10 experimental days (*Figure 4C* top). The rate of inducing an electrographic event was defined to be the percentage of times the activation stimulus induced afterdischarges lasting a minimum of 5 s. A two-sided pairwise t test determined that, on all 10 days, the rate of inducing an electrographic event did not significantly

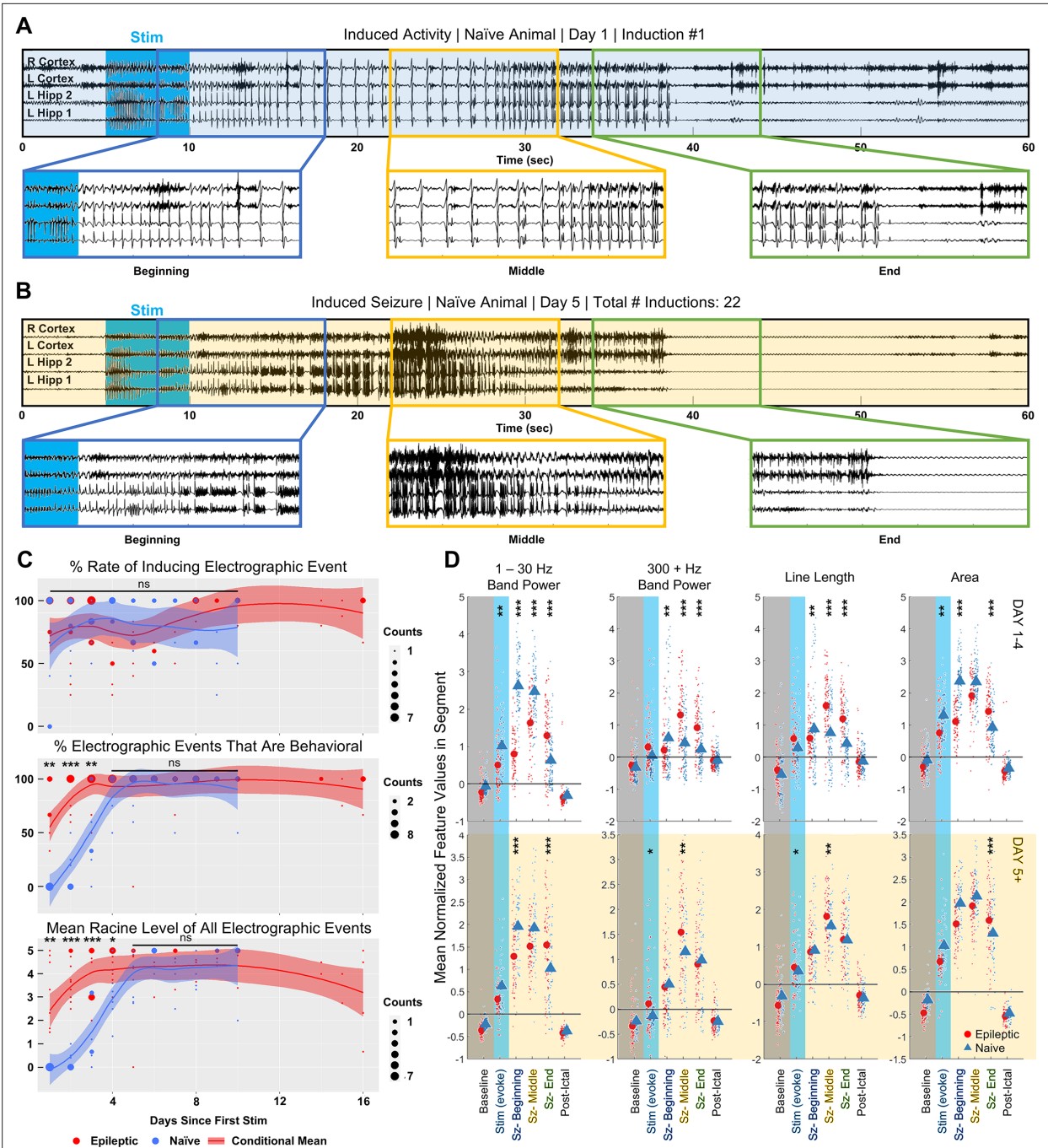

**Figure 4.** Inducing seizures in epileptic animals differs from optical kindling in naive animals. (**A**) Initial optogenetic activations in naive animals induced low frequency afterdischarges. Representative EEG trace with three segments of 10 s at the start, middle, and end of activity enlarged. (**B**) Following multiple days of stimulation, application of activation stimulus in the same animal induced Racine 5 seizures. (**C**) Rate of electrographic afterdischarges from optogenetic activation did not significantly differ between naive and epileptic animals. Percent of behavioral electrographic events significantly differed between naive and epileptic animals on stimulation day 1 through 3. Average Racine score of electrographic inductions significantly differed between naive and epileptic animals on stimulation day 1 through 4. Pairwise T test: * p<0.05, ** p<0.01, *** p<0.001. (**D**) In the first 4 days of stimulation, inductions in naive (100 inductions, n=7) and epileptic animals (87 inductions, n=10) significantly differed in the feature space. Differences were reduced, but still existed, on stimulation day 5 or later (epileptic – 75 inductions, n=7; naive – 58 inductions, n=7). Linear Mixed Effect Model: * p<0.05, ** p<0.01, *** p<0.001.

The online version of this article includes the following figure supplement(s) for figure 4:

**Figure supplement 1.** Examples of inductions in naive animal #110.

**Table 1.** Behavioral scoring and Racine Scale assignments.

| Racine | Behaviors |
| --- | --- |
| 0 | Normal behavior, grooming, free movement (even with slight limp) |
| 1 | Clear freezing/flattening of the body |
| 2 | Tail stiffening/forelimb clonus/uncontrolled shaking |
| 3 | Rearing |
| 4 | Wild run/backpedaling |
| 5 | Uncontrolled jumping/loss of righting reflex |

differ between the epileptic and naive animals (p>0.05). After integrating data across all stimulation days, it was determined that the overall rate of inducing an electrographic event in naive animals was approximately 86.0 ± 6.0%, which also did not significantly differ from that of the epileptic animals (p=0.8506, *Figure 3—figure supplement 1*). However, the average duration of induced activity in naive animals was 20.87±2.19 s, which was significantly shorter than the average duration of induced activity in epileptic animals (30.98±4.69 s, p=0.0005, *Figure 3—figure supplement 1*).

We next analyzed the percentage of electrographic events that exhibited a behavioral manifestation. This comparison between naive and epileptic animals included behaviors such as visible forelimb clonus, uncontrolled shaking, tail stiffening, uncontrolled rearing, wild running, loss of righting reflex, and jumping (*Figure 4C* middle, *Table 1*). We did not include behaviors in which the animal retained control over its body, such as exploring or grooming. A two-sided pairwise t test found that the percentage of electrographic events with a behavioral seizure manifestation significantly differed between the naive and epileptic animals on the first (p=0.0056), second (p<$1e^{-4}$), and third (p=0.0088) stimulation day. No significant difference in behavioral seizure manifestations was found on stimulation day 4 or later (p>0.05). To further quantify the behavioral differences, we compared the Racine seizure score for the induced activity (*Figure 4C* bottom). A two-sided pairwise t test found that the average Racine seizure score significantly differed between the naive and the epileptic animals on the first (p=0.0087), second (p<$1e^{-4}$), third (p<$1e^{-4}$), and fourth (p=0.024) stimulation day. No significant difference in Racine seizure score was found on stimulation day 5 or later (p>0.05).

Later activations and their differing behavior indicated that the animal entered a distinct state after multiple days of optical activation. Thus, we decided to perform separate feature space comparisons of the electrographic activity between naive animals and epileptic animals for the initial activations (days 1–4) and the later activations (day 5 +; *Figure 4D*). To standardize the comparison between naive and epileptic induced activity, we followed a similar procedure to the analysis did in *Figure 3*. We used an interactive linear mixed effect model that accounts for the intra-animal variability to uncover the inter-animal changes in electrographic feature over three segments. The terciles of the 308 inductions in naive animals were computationally determined by the same k-nearest neighbor classifier used in epileptic animals. To reduce noise from shorter events, only induced activity lasting more than 15 s was compared. The initial activations group was composed of 100 events from 7 naive animals and 87 events from 10 epileptic animals. The later activations group was composed of 75 events from 7 epileptic animals and 58 events from 7 naive animals.

Consistent with visual observation of initial activations, increases in the electrographic features versus baseline significantly differed between naive and epileptic animals (*Figure 4D* top). Overall, naive-induced activity tended to have higher area and more low-frequency band power while epileptic induced activity tended to have more high-frequency band power and higher signal complexity. The linear mixed effect model found significant differences in the increase of 1–30 Hz band power from baseline to the beginning third (p<$1e^{-4}$), middle third (p<$1e^{-4}$), and final third (p<$1e^{-4}$). The difference in the 300–1000 Hz band power from baseline was also significant between the two groups at the beginning third (p=0.002), middle third (p<$1e^{-4}$), and final third (p<$1e^{-4}$). A similar case existed for line length at the beginning third (p=0.008), middle third (p<$1e^{-4}$), and final third (p<$1e^{-4}$) and for area at the beginning third (p<$1e^{-4}$) and final third (p<$1e^{-4}$). The increase in area from baseline to the middle third was barely insignificant (p=0.059).

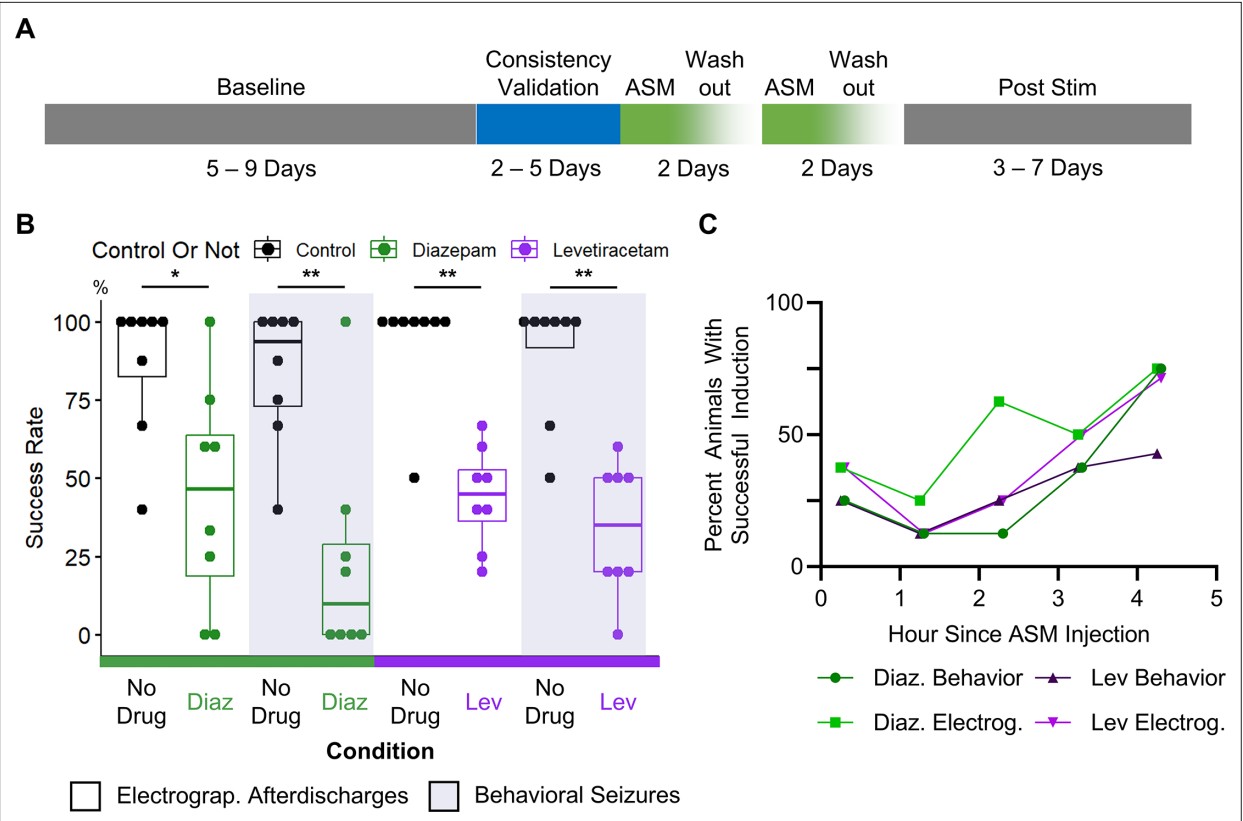

**Figure 5.** Induced seizures in epileptic animals responded to both diazepam and levetiracetam. (**A**) Experimental timeline for testing ASM efficacy in on-demand seizure model. Stimulus for seizure induction was tested for consistency before ASM application. A 48 hr washout occurred between subsequent ASM applications. (**B**) Post ASM application, rates of inducing electrographic afterdischarges (any electrographic activity lasting more than five seconds) and behavioral seizures (generalized seizures with seizure behaviors) were significantly reduced. Paired One-Tailed Wilcoxon Signed Rank Test, * p<0.05, ** p<0.01 (**C**) Probability of successful induction of activity (averaged across all epileptic animals) increased the more time has passed since ASM injection into the mouse.

The online version of this article includes the following figure supplement(s) for figure 5:

**Figure supplement 1.** Racine level of seizures, measured as percentage of all inductions in epileptic animals before and after drug application.

Differences in the feature space were reduced in later activations (*Figure 4D* bottom). The differences in the increase of the 1–30 Hz band power between the two groups were reduced in the beginning third (p=0.0002), middle third (p=0.051), and final third (p<1e$^{-4}$). The same applied to 300–1000 Hz band power (beginning third p=0.60, middle third p=0.029, final third p=0.097), line length (beginning third p=0.21, middle third p=0.002, final third p=0.11), and area (beginning third p=0.15, middle third p=0.56, final third p<1e$^{-4}$). Despite reductions between feature values in the later activations group, not all significant differences between the naive and epileptic animals were eliminated. The persistent differences highlighted by the linear mixed effect model show the importance of evaluating novel ASMs in epileptic animals.

## Common anti-seizure medications could suppress induction of electrographic and behavioral seizures

To determine the utility of the Opto-IHK model for evaluating ASMs, we tested whether FDA-approved and commonly prescribed ASMs, such as diazepam and levetiracetam (*Glauser et al., 2016*), could reduce the rates of both electrographic afterdischarge induction (electrographic activity lasting more than 5 s, including seizures) and behavioral seizure (generalized seizure with behavioral component) induction in epileptic animals. Diazepam, a benzodiazepine that interacts with GABA receptors, is commonly prescribed for first-line control of convulsive seizures (*Segal et al., 2021*),

whereas levetiracetam is an anticonvulsant that reduces overall network excitability through effects mediated by the synaptic protein SV2A (*Contreras-García et al., 2022*).

The experimental paradigm is shown in *Figure 5A*. Prior to inducing activity with the optogenetic stimulus, the baseline spontaneous seizure frequency was recorded in nine epileptic animals. The next 2–5 days were used for determining and validating the consistency of the threshold stimulus. On each day of ASM testing, optogenetic activation was performed hourly to establish the drug-free electrographic discharge induction and behavioral seizure induction success rate. Then, the animal received either a single intraperitoneal injection of 800 mg/kg levetiracetam or a single subcutaneous injection of 5 mg/kg of diazepam. The activation stimulus was applied to the animal approximately 10 min after ASM administration and was re-applied hourly. A washout period of 48 hr, if applicable, occurred between subsequent ASM administrations. After completion of drug testing, animals underwent a post-stimulation recording period of up to 7 days.

Application of diazepam significantly reduced the rates of electrographic discharge and behavioral seizure induction in eight assessments conducted on six epileptic animals (*Figure 5B*). Typically, after diazepam administration, optogenetic activation did not induce electrographic activity. Even when afterdischarges were successfully induced, animals typically did not display seizure-like behaviors. In all animals, the severity of behavioral seizures is reduced after diazepam administration (*Figure 5—figure supplement 1*). Integrating data from all animals, prior to drug administration, electrographic discharge induction rate averaged 86.77 ± 22.3% and the behavioral seizure induction rate averaged 83.65 ± 21.9%. The paired one-tailed Wilcoxon signed rank test found that after drug administration, the electrographic discharge induction rate was significantly reduced to 44.17 ± 35.8% (p=0.016) and the behavioral seizure induction rate was significantly reduced to 23.12 ± 34.5% (p=0.008). One epileptic animal was a diazepam non-responder on one of two testing days. Excluding the diazepam non-responder, no tonic-clonic seizures (Racine >3) were induced 3 hr after drug administration in any of the five epileptic animals. Even in the diazepam non-responder, the Racine scale of induced seizures was lower after diazepam administration (*Figure 5—figure supplement 1*).

Application of levetiracetam significantly reduced the rates of electrographic discharge and behavioral seizure induction in eight assessments on five epileptic animals (*Figure 5B*, *Figure 5—figure supplement 1*). After levetiracetam administration, electrographic inductions were typically unsuccessful. In mice, levetiracetam has a short half-life of approximately 3.2 hr (*Song et al., 2018*). In our study approximately 4 hr after the injection, tonic-clonic behavioral seizures (Racine >3) were successfully induced in two out of five epileptic animals. Integrating data from all animals, prior to drug administration, average electrographic discharge induction rate was 93.75 ± 17.7% and the average behavioral seizure induction rate was 89.58 ± 19.8%. A paired one-tailed Wilcoxon signed sum test found that after drug administration, electrographic discharge induction rate was significantly reduced to 43.96 ± 16.1% (p=0.004), and the behavioral seizure induction rate was significantly reduced to 33.75 ± 21.3% (p=0.004). Additionally, in all animals, seizure burden, measured as the severity of behavioral seizures decreased after levetiracetam administration (*Figure 5—figure supplement 1*).

Quantifying the seizure induction rates after drug administration by hour showed that both drugs had maximal effect between the first and the second hour (*Figure 5C*). As the drug was metabolized or excreted, the rate of successful seizure induction increased, eventually approximating 75% of all animals by the fourth hour. This was expected, as seizure induction rates should gradually return to pre-drug levels as the drug was cleared from the animal body. Taken together, these results showed that the Opto-IHK model has potential for evaluating ASM efficacy in preventing seizures and guiding calculations on ASM pharmacokinetics.

## Discussion

In this study, we showed that selective activation of CA1 principal cells could induce on-demand seizures in both chronically epileptic and naive animals. Induced seizures were reliably generated, reproducible, and bore striking similarities to spontaneously occurring seizures. Induced activity differed between epileptic animals and naive animals, indicating that the epileptic network is critically different from that of the healthy brain. Finally, induced seizures could be suppressed with standard ASMs, proving the Opto-IHK model has utility for assessing the effectiveness of potential therapies.

## Seizure induction takes advantage of mechanistic changes in the hippocampus

The seizure induction procedure took advantage of naturally occurring changes in the epileptic brain to elicit seizures on demand, suggesting that hyperexcitation of principal cells could be a mechanism by which activity propagates out from the hippocampus and generalizes into behavioral seizures. The profound difference in the induced activity between naive and epileptic animals further suggested that the epileptic circuit is a key facilitator of seizure generalization – only in the epileptic circuit could we provoke behavioral seizures from stimulation day 1.

Previous research suggests that changes in the CA1 during epileptogenesis contribute to increased bursting activity in chronically epileptic animals but stops short of establishing a causal relationship between hyperactivity and generalized seizures (*Meier and Dudek, 1996*; *Chen et al., 2011*; *Shao and Dudek, 2004*; *Perez et al., 1996*). In the pilocarpine model of chronic epilepsy, CA1 principal cells in epileptic animals were prone to firing synchronous bursts of spikes when excited with inputs that induced only single spikes in naive animals (*Sanabria et al., 2001*). Inputs to the CA1 also change during epileptogenesis. The temporoammonic pathway, a direct connection from the entorhinal cortex to the CA1, switched from being a highly regulated, weak excitatory input in naive animals to a powerful excitatory pathway in epileptic animals (*Aksoy-Aksel and Manahan-Vaughan, 2013*; *Ang et al., 2006*). Electrical activation of the temporoammonic pathway was also sufficient to generate bursting activity in chronically epileptic brain slices (*Ang et al., 2006*). Despite these pioneering studies, there was no direct investigation into whether seizures in the epileptic brain can initiate purely from CA1 hyperexcitation. In our work, we answer this question by showing that we can harness CA1 principal cells to induce generalized seizures on demand. We also show that these seizures are similar to spontaneous seizures in the EEG feature space; further suggesting that, in epileptic animals, this pathway is potentially active in seizure generation.

Behaviorally, induced seizures resembled tonic-clonic seizures in human patients (*Theodore et al., 1994*). Induced seizures were characterized by an initial tonic phase, where the dominant behavior was freezing, and the EEG showed continuous fast spiking activity. The tonic phase is followed by a clonic phase, where the animal displayed uncontrolled movements, such as backpedaling, rearing, and jumping. During the clonic phase, we observed clearly defined bursting and large amplitude single spikes. The gross similarity in the EEG and the behavior posits a question – could CA1 hyperexcitation be one of the mechanisms by which tonic-clonic seizures naturally start? Further studies with the Opto-IHK model could answer questions about the initiation and the termination of such seizures, with clinical implications in both the epidemiology and the treatment of medication-resistant seizures.

## Application of model to testing multiple classes of ASMs, closed-loop neuromodulation therapies, and studying cellular interactions

In this work, we found that seizures induced in epileptic animals could be blocked by applying two commonly used ASMs – diazepam and levetiracetam. These two medications were chosen because they act through different mechanisms – diazepam enhances GABA receptor efficacy and levetiracetam disrupts synaptic transmission (*Segal et al., 2021*; *Contreras-García et al., 2022*). Diazepam historically performs well in many seizure models. However, levetiracetam is unique among many ASMs in that it was originally missed by the standard ASM screens of the maximal electroshock seizure and subcutaneous pentylenetetrazol seizure tests. Its antiepileptic effect was only discovered using a kindling model of epilepsy, which has since been integrated into the ETSP pipeline (*Löscher et al., 1998*; *Löscher and Hönack, 1993*). In the Opto-IHK model, levetiracetam had a clear antiseizure effect. We believe that the etiological relevance of our model allows levetiracetam to exert its effect on synaptic transmission and prevent seizure generalization. These experiments show that the Opto-IHK model could be used for testing multiple classes of ASMs.

Outside of pharmacologics, the Opto-IHK model could be applied to evaluating time-dependent, closed-loop treatment paradigms. The model allows for precise timing of seizure onset; thus, experimenters could rapidly test the effects of both open and closed-loop electrical stimulation on seizure initiation, propagation, and termination. We envision the Opto-IHK model to enable quick advances in determining optimal targets and tuning parameters for closed-loop stimulation. There is an urgent need to explore the parameter space for closed-loop seizure control, as the multitude of options makes it difficult to quickly optimize responsive neurostimulators for maximal seizure control

(*Sisterson et al., 2020*). A reliable, physiologically realistic on-demand seizure model could dramatically accelerate this process.

Alternatively, our approach can be harnessed to answer questions of biological and clinical significance. For instance, adaptation of our approach into rare disease models of epilepsy, such as the Scn1a mouse model used to study Dravet Syndrome (*Escayg et al., 2000*), could produce fresh insights into how seizures spread in different models. Comparing seizure propagation across various models will also help elucidate how genetic mutations and cellular abnormalities influence seizure mechanisms across the heterogeneity of human epilepsy. Furthermore, circuit-specific investigations can be conducted by integrating our model with cell-specific optogenetic or chemogenetic approaches (*Jamiolkowski et al., 2024*). By evaluating seizure induction rate while suppressing or activating certain neural populations, one can quickly assess the potential of the circuit-specific intervention strategy to control seizures.

Our experiments suggest that interfering with activity from the CA1 region may be one way to stop seizure progression. Diazepam and levetiracetam both reduced the likelihood of induced activity generalizing beyond the hippocampus, resulting in lower rates of electrographic activity induction and lower rates of behavioral seizure induction. As the CA1 is the main output of the hippocampus, deciphering ways to prevent hippocampal activity from generalizing could be impactful for clinical care. If abnormal hippocampal outputs could not leave the hippocampus, patients with seizures that originate in the hippocampus might not experience the debilitating effects of grand mal seizures.

## Comparison to other seizure models used in pharmacologic screening

Other models that acutely induce seizures exist, but they either do not use chronically epileptic animals or do not have as rapid onset as the one presented in this work (*Paschen et al., 2020*). Some models utilize a kindling paradigm, requiring multiple days of successive activation before seizures can be reliably induced (*Chen et al., 2016*). Other approaches are performed under anesthesia (*Wagner et al., 2015*) or utilize naive animals (*Mueller et al., 2023*), both of which could introduce differences in neural activity when compared to studies in freely moving epileptic animals. Overall, prior studies we reviewed do not sufficiently address the challenges of efficiently evaluating epilepsy treatments in etiologically relevant models.

Conversely, the Opto-IHK model allows for more robust powering of studies in etiologically relevant models. We target a potential seizure mechanism in the CA1 to reliably elicit seizures hourly in freely moving animals. Our model also generates both long afterdischarges and tonic-clonic behavioral seizures. This is unlike models that rely on continuous stimulation throughout the seizure to generate behavioral events or models that pre-screen for animals that are predisposed to have spontaneous seizures (*Mueller et al., 2023*; *Grabenstatter and Dudek, 2019*). In our model, all animals are used, and ictal activity naturally propagates out of the hippocampus and evolves into generalized tonic-clonic seizures. The Opto-IHK model is also unlike traditional kindling models because we target specific cellular populations instead of using non-specific electrical stimulus (*Chen et al., 2016*). Thus, seizures from the Opto-IHK model are likely to be highly relevant for testing clinical treatments and developing novel approaches for epilepsy.

Long term, we envision the Opto-IHK could be integrated into a drug screening pipeline such as the Epilepsy Therapy Screening Program (ETSP) (*Wilcox et al., 2020*). Currently, the ETSP utilizes a mixture of models, including acutely induced seizures in naive animals and spontaneous recurrent seizures in chronically epileptic, IHK rodents. An intermediate step in the ETSP evaluates how ASMs affect hippocampal paroxysmal discharges (HPDs), which are frequent non-behavioral, focal electrographic seizures, to quickly produce dose-response curves with high confidence (*Duveau et al., 2016*). However, HPDs are not generalizable across species – they are specific to the mouse model (*Klee et al., 2017*). In addition, it is unclear whether medications that prevent HPDs would also be effective against generalized convulsive seizures or partial non-convulsive seizures in humans.

In line with modern medicine's aim to develop etiology-specific drugs as part of a precision medicine approach, it is crucial to incorporate evaluation of convulsive seizures into the screening process to discover drugs that can effectively stop these seizures. The Opto-IHK model is well positioned to supplant HPDs in the existing pipeline. First, on-demand seizures do not require more time to prepare than HPDs. In the preparation phase, both methods require chronic epilepsy induction and an EEG implantation surgery. In the screening phase, on-demand seizures produce significant time savings,

as seizures can be evaluated immediately instead of waiting for HPD characterization and analysis. Furthermore, as on-demand seizures are behavioral seizures, we predict that our model will respond to more categories of ASMs than HPDs, serving as a quick initial check as to whether the drugs' actions are sustained on seizures in the epileptic brain. Some steps are still needed prior to integration - additional scaling of the Opto-IHK model and validation with more ASMs are critical further studies. These experiments could better characterize the model and help determine whether the Opto-IHK can serve as a testbed for pharmacologics that specifically target refractory, drug-resistant seizures.

In all, we present the Opto-IHK as a biologically relevant, higher throughput on-demand seizure model that can be used to evaluate novel pharmacologics and time-sensitive treatment paradigms. The Opto-IHK takes advantage of a biologically relevant mechanism to generate long afterdischarges and behavioral tonic-clonic seizures. With this model, we envision immediately increasing the speed of discovering and evaluating new clinical treatments for epilepsy and the identification of new mechanisms behind the initiation of tonic-clonic seizures.

# Materials and methods

**Key resources table**

| Reagent type (species) or resource | Designation | Source or reference | Identifiers | Additional information |
|---|---|---|---|---|
| Genetic Reagent (*Mus musculus*) | Thy1-ChR2-YFP | Jackson Laboratory (JAX) | IMSR_JAX:007612 | |
| Chemical compound | Kainic Acid | Hello Bio | HB0355 | |
| Chemical compound, drug | Diazepam | Dash Pharmaceuticals | 69339013634 | 5 mg/ml |
| Chemical compound, drug | Levetiracetam | Sigma Aldrich | PHR-1447 | |
| Other | Optic Fiber Cannula | RWD | 907-03011-00 | 1.25 mm ferrule, 400 micron core, 0.39 NA |
| Other | Optic Fiber (FC-PC) | RWD | 807-00059-00 | 200 micron core, 0.22 NA |
| Other | Optic Fiber (FC-FC) | Doric | D202-2075 | 200 micron core, 0.22 NA |
| Other | Laser Power Source | Laserglow | LRS-0473 | |
| Chemical compound, drug | Cresyl Violet Acetate | Sigma Aldrich | C5042 | For Staining |
| Software, algorithm | MATLAB | MATLAB | RRID:SCR_001622 | 2019b |
| Software, algorithm | R Project for Statistical Computing | R | RRID:SCR_001905 | 4.4.0 |

## Study design

The purpose of this study was to create a higher throughput model for evaluating ASMs in the diseased, epileptic brain. To do so, we tested whether specific optical activation of CA1 principal cells could induce generalized behavioral seizures in epileptic animals. Next, we compared these induced seizures to spontaneous seizures to establish the model's face validity. To show that successful induction was dependent on changes in the epileptic brain, we attempted to provoke seizures in naive animals. Comparing the induced activity between naive and epileptic animals allowed us to establish the model's construct validity. Finally, we attempted to use known ASMs to stop seizures. Diazepam and levetiracetam reduced the likelihood of seizure induction, establishing that our model has predictive validity. Blinding was not utilized in this study.

## Animal experiments

All procedures are approved in accordance with the lab's Institutional Animal Care and Use Committee (IACUC) protocol at the Children's Hospital of Philadelphia, which is an Association for Assessment and Accreditation of Laboratory Animal Care International (AAALAC) accredited site. The study protocols are IAC 446 and IAC 1113.

## Epilepsy induction

C57BL/6J-Thy1-ChR2-YFP mice (RRID:IMSR_JAX:007612) underwent intrahippocampal injection of 50 nL, 20 mM kainic acid (Hello Bio) under anesthesia (coordinates: right CA3 AP – 2.7, ML +3.0, DV

– 3.2, right CA1 AP – 2.0, ML +1.6, DV – 1.6). After the onset of status epilepticus, 5 mg/kg diazepam was administered to reduce the severity of seizures. Animals that died from epilepsy induction were excluded.

## EEG and fiber implantation

Animals were implanted with a six-channel electrode pedestal (P1 Technologies) to two cortical screws (AP +0.5, ML ±1.7), two cerebellar screws (AP – 5.5, ML ±2.0, ground and reference), and a pair of twisted hippocampal depth wires (AP – 2.0, ML – 1.7, DV – 1.6). A 400 µm diameter optic fiber cannula (RWD Life Science) was positioned at 2.0 mm (AP – 2.0, ML +1.7) depth illuminating the damaged CA1. Dental cement (Lang Dental) secured the entire apparatus.

## Video EEG and optogenetic stimulus

Animals were placed into a Plexiglass cage and connected to a Stellate Harmonie (Natus) recording system. A Master 8 controller that was set to 10 Hz, 25 ms pulses was used to control the delivery of light from a Laserglow 473 nm laser. Laser power and stimulation duration depicted in *Figure 3— figure supplement 1*. Stimulation was performed every 1–3 hr over many days. Animals that had no light response at all – not even a stimulation artifact – were excluded. During the entire recording, water and food were provided ad libitum. Mice were in a 12 hr light/dark cycle.

## Pharmaceuticals

On each testing day, a few optogenetic activations occurred prior to drug injection to determine the pre-drug induction success rate. Animals then received either one subcutaneous injection of 5 mg/kg diazepam (Dash Pharmaceuticals) or an intraperitoneal injection of 800 mg/kg levetiracetam (Sigma-Aldrich). Up to six more activations occurred every 1–1.5 hr following ASM administration. After the experiment, animals had a break of 1–2 days during which there was no optical stimulation. A one-tailed Wilcoxon matched pairs signed rank test was used to compare the success rate before and after ASM administration.

## Perfusion and histology

Animals were deeply anesthetized and transcardially perfused with PBS and fixed in 4% paraformaldehyde in PBS. Brains were then placed into 30% sucrose in PBS for cryoprotection. Cryoprotected brains were sliced at 50 µm thickness in the coronal plane on a microtome. Slices were stained with 0.1% cresyl violet (Sigma Aldrich) using the Nissl staining protocol. Briefly, slices were dehydrated in increasing concentrations of reagent alcohol, beginning with 75%, then progressing to 95%, and ending with 100% reagent alcohol. Next, slices were demyelinated with xylene (Sigma-Aldrich). After demyelination, slices were rehydrated with decreasing concentrations of reagent alcohol, beginning with 100%, then progressing to 95%, 75%, and 50% reagent alcohol. The final step of the rehydration is an incubation in DI water. After rehydrating, slices were stained in 0.1% cresyl violet in DI water for 20–40 min at room temperature. Slices were then rinsed in DI water, before being dehydrated and destained with fresh 50%, 75%, 95%, and 100% reagent alcohol. For the final cleaning, slices were rinsed in xylene. DPX mounting media (Sigma-Aldrich) was used to mount the slices, and the slides were imaged on a Leica DMIRB microscope using the LasX software.

## Statistical analysis and computational processing

All code is available on Github at the following repository: https://github.com/yuzhangc/Evoked_ Seizures (copy archived at *Chen, 2025*).

### Preprocessing

Recorded EEG data was filtered with a second-order IIR notch filter to remove 60 Hz line noise and then z-score normalized to a baseline period – defined as 5 s immediately preceding the onset of the optogenetic stimulus. A 4 Hz sixth-order Butterworth high-pass filter was used to remove low-frequency artifacts.

## Feature calculation

The following features were extracted from 500 ms windows with 250 ms displacement. In the below formulas, $N$ represents the number of items in the recording, indexed by $n$, and $x$ is an element in the recording.

1. Line length (*Esteller et al., 2001*), which measures the complexity of the signal: $\sum_{n=2}^{N} |x_n - x_{n-1}|$
2. Area (*Fergus et al., 2016*) $\sum_{n=1}^{N} |x_n|$
3. Energy $\sum_{n=1}^{N} x_n^2$
4. Zero crossings around mean (*Pyrzowski et al., 2021*)
   $$\sum_{n=2}^{N} \mathbf{1}\big((x_{n-1} - \bar{x} > 0 \wedge x_{n-1} - \bar{x} < 0) \mathbf{I}(x_{n-1} - \bar{x} < 0 \wedge x_{n-1} - \bar{x} > 0)\big)$$
5. Root mean squared amplitude $\sqrt{\frac{1}{N} \sum_{n=1}^{N} |x_n|^2}$
6. Skewness (*Hosseini et al., 2018*) $\dfrac{\frac{1}{N} \sum_{n=1}^{N} (x_n - \bar{x})^3}{\left(\sqrt{\frac{1}{N} \sum_{n=1}^{N} (x_n - \bar{x})^2}\right)^3}$
7. Approximate entropy using the MATLAB approximateEntropy function.
8. Lyapunov exponent, a measure of divergence, using the MATLAB lyapunovExponent function.
9. Phase locked high gamma (*Weiss et al., 2015*). Data was filtered into LFP (4–30 Hz) and high gamma (80–150 Hz) portions using a second-order Butterworth filter. A Hilbert transform (H) was used to extract imaginary and real components from both LFP (*lfp*) and high gamma (*hg*) data. The LFP phase ϕ was calculated using the two-argument archtangent. The high gamma amplitude (A) was calculated by taking the square root of the sum of the following: (1) the square of the imaginary component of the high gamma data and (2) the square of the real component of the high gamma data. The phase locked high gamma is calculated as the absolute value of the mean of the following product: $\left| H\left(hg\right) \right| \times e^{i\left(\phi\left(lfp\right) - A\left(hg\right)\right)}$.
10. Magnitude squared coherence between channels (*Gao et al., 2023*) was calculated using the MATLAB mscohere function. Channels to calculate coherence with were defined to be between the set of hippocampal wires (Channels 1 and 2), the two EEG screws (Channels 3 and 4), the ipsilateral screw and wire (Channels 1 and 3), and the contralateral screw and wire (Channels 1 and 4). The function calculated the power spectral density (P) using the Welch's overlapped periodogram function. The function also calculated the cross power spectral density of the two channels' data (x). As an example, the magnitude squared coherence (C) for channels 1 and 2 was $C_{1,2}\left(x\right) = \dfrac{\left|P_{1,2}\left(x\right)\right|^2}{P_{1,1}\left(x\right) P_{2,2}\left(x\right)}$
11. Mean absolute deviation $\frac{1}{N} \sum_{n=1}^{N} |x_n - \bar{x}|$
12. Band power between 1–30 Hz, 30–300 Hz, and 300+ Hz. Band power was calculated using the MATLAB bandpower function. The function generated a power spectral density estimate using the Hamming window and integrated the area from the periodogram between the frequencies of interest.

After calculation, individual features were z-scored normalized by subtracting the mean of the feature values for the entire sequence from the feature values and dividing by the standard deviation of the feature values for the entire sequence.

## Induced activity length determination, thresholds, and calculation of thirds

An unsupervised K nearest neighbor model was trained with z-score normalized input from one spontaneous seizure. The model had three output classes – two were classified as 'seizure' and the last one was 'baseline'. The model was then used to determine induced activity length across all events from all animals, with manual adjustment for about 20% of all seizures. Event durations were averaged per animal, and a two-tailed Wilcoxon rank sum test was used to compare the average of the naive animals to the average of the epileptic animals. Overall induction success rate was also compared with a two-tailed Wilcoxon rank sum test between naive and epileptic animals. However, daily activation success rate (by day since first stimulation) was compared using a two-sided pairwise t test.

The threshold for activity induction was defined as the minimum duration and laser power at which a 10 Hz optical stimulus would cause a minimum 5-s-long electrographic activity at least 66.67% of the time. Only drug-free conditions were considered. A two-tailed Wilcoxon rank sum test was used

to compare the threshold duration and the threshold power distribution of the naive and epileptic animals.

Features calculated in Section B were then segregated into 1 of 6 time points based on the length of induced activity: (1) before stimulation, (2) during stimulation, (3) initial/beginning third, (4) second/middle third, (5) final/ending third, and (6) post-stimulation (30 s). Linear mixed effect models (using the lmer function in the lmerTest package) were used to determine how the comparison (such as epileptic vs naive or spontaneous vs induced) influenced the EEG features over time points.

### Spontaneous seizure detection and video EEG behavioral scoring

Spontaneous seizures were detected using custom-written MATLAB code (*Kahn et al., 2019*). Comparisons on daily spontaneous seizure frequency used a two-tailed Mann Whitney test. Uncontrolled behaviors and the Racine scale (*Racine, 1972*) associated with the behaviors are as follows and described in *Table 1*.

The Racine behavior score of the event was determined to be the highest value out of all the behaviors the animal displayed during each event. If there was no uncontrolled behavior displayed, or if the animal did not move at all throughout the entire event, the score was 0 – no change.

The mean Racine level of induced activity was compared between naive and epileptic animals by a two-sided pairwise t test. A two-sided pairwise t test was also used to compare the rate of electrographic events that had a behavioral manifestation (a score above 0).

### Linear mixed effect models and support vector machines

Mixed effects models were fitted with the lmer function in the lmerTest package in R. Feature values calculated in Section B and segregated in Section C were exported from MATLAB to R. The baseline was set to be the pre-stimulation values for each feature. Since we wanted to compare the extent to which features changed from pre-stimulation values to each of the event terciles, the time point was a fixed effect in all mixed effect models. The animal was set as the random effect due to differences in the induced activity between animals.

The other fixed effect in the mixed effect model depended on what groups we were comparing. Between the epileptic and naive animals, the fixed effect was whether the animal was epileptic or not. Between the spontaneous seizures and induced activity, the fixed effect was whether there was an optogenetic stimulus. The mixed effect model equation was thus: EEG.Feature~Comparison * Time. Point + (1 | Animal).

As mentioned in the main text, additional filtering of events occurred before they were fed into the mixed effect model. Naive events differed significantly between early and late activation days, so for the epileptic versus naive induction comparisons, events were split by the day since stimulation start (days 1–4 and day 5+). The spontaneous seizure analyzer (Section E) could only identify behavioral seizures with Racine score 3 or above. Thus, for the spontaneous seizures versus induced activity comparison in epileptic animals, only events with Racine score of 3 or above were compared. For both comparisons, events shorter than 15 s were excluded to reduce noise from shorter afterdischarges. All drug treatment trials were also removed.

A new support vector machine for classifying induced activity was trained per animal. First, features were calculated from ten baseline EEG segments from before stimulation began. Next, these features were combined with the features from the animal's spontaneous seizures to form the training inputs. The SVM algorithm was trained using that input set on separating events as either 'spontaneous seizure' or 'baseline'. The testing data consisted of features from all optical activations. The model was asked to classify the induced activity. Accuracy and error rates were calculated based on ground truth, where a successful induction was defined as an electrographic event with afterdischarges lasting a minimum of 5 s.

## Acknowledgements

We acknowledge Dr. Douglas Coulter for his invaluable scientific insights and support on our project. We acknowledge the support of Alicia White and Emily Schellinger for their technical assistance in this project, as well as Dr. Srdjan Joksimovic and Dr. Anthoni Goodman for their helpful suggestions. We also acknowledge Dr. Mary Putt, Director of the IDDRC Biostatistics and Data

Science Core, for her statistical advice and consultation. NIH NINDS 5-T32-NS-091006-07 (BL, YC), NIH 1P50HD105354 (IDDRC at CHOP/Penn), NIH R01NS082046 (DC, HT), NIH 1DP1 NS122038-01 (BL), Mirowski Family Foundation (BL), Neil and Barbara Smit (BL), Jonathan and Bonnie Rothberg (BL), CHOP AEF (HT).

## Additional information

### Funding

| Funder | Grant reference number | Author |
|---|---|---|
| National Institutes of Health | NS-091006-07 | Yuzhang Chen Brian Litt |
| National Institutes of Health | P50HD105354 | Hajime Takano |
| National Institutes of Health | R01NS082046 | Hajime Takano |
| National Institutes of Health | NS122038-01 | Brian Litt |
| Mirowski Family Foundation | | Brian Litt |
| Neil and Barbara Smit | | Brian Litt |
| Jonathan and Bonnie Rothberg | | Brian Litt |
| CHOP AEF | | Hajime Takano |

The funders had no role in study design, data collection and interpretation, or the decision to submit the work for publication.

### Author contributions

Yuzhang Chen, Conceptualization, Data curation, Software, Formal analysis, Validation, Investigation, Methodology, Writing - original draft; Brian Litt, Supervision, Funding acquisition, Writing – review and editing; Flavia Vitale, Supervision, Writing – review and editing; Hajime Takano, Conceptualization, Data curation, Supervision, Funding acquisition, Investigation, Methodology, Project administration, Writing – review and editing

### Author ORCIDs

Yuzhang Chen ⓘD https://orcid.org/0000-0002-7377-861X
Flavia Vitale ⓘD https://orcid.org/0000-0001-8644-550X
Hajime Takano ⓘD https://orcid.org/0000-0003-3033-2412

### Ethics

All procedures are approved in accordance with the lab's Institutional Animal Care and Use Committee (IACUC) protocols (#1113 and #446) at the Children's Hospital of Philadelphia.

Reviewer #1 (Public review): https://doi.org/10.7554/eLife.101859.3.sa1
Reviewer #2 (Public review): https://doi.org/10.7554/eLife.101859.3.sa2
Reviewer #3 (Public review): https://doi.org/10.7554/eLife.101859.3.sa3
Author response https://doi.org/10.7554/eLife.101859.3.sa4

## Additional files

### Supplementary files

MDAR checklist

## Data availability

All data have been deposited at https://doi.org/10.26275/4uue-pck4 and are publicly available. The code for the analyses presented in this paper is also accessible at https://github.com/yuzhangc/Evoked_Seizures (copy archived at *Chen, 2025*).

The following dataset was generated:

| Author(s) | Year | Dataset title | Dataset URL | Database and Identifier |
|---|---|---|---|---|
| Takano H, Vitale F | 2025 | On-Demand Seizures Facilitate Rapid Screening of Therapeutics for Epilepsy | https://doi.org/10.26275/4uue-pck4 | Pennsieve Platform, 10.26275/4uue-pck4 |

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
