## [Editor Report · eLife Assessment]

The authors modified a common method to induce epilepsy in mice to provide an improved approach to screen new drugs for epilepsy. This is **important** because of the need to develop new drugs for patients who are refractory to current medications. The authors' method evokes seizures to circumvent a low rate of spontaneous seizures and the approach was validated using two common anti-seizure medications. The strength of evidence was **solid**, making the study invaluable, but there were some limitations to the approach and methods.

---

## [Referee Report · Reviewer #1 (Public review)]

Summary:

This important study by Chen et. al. describes a novel approach for optogentically evoking seizures in an etiologically relevant mouse model of epilepsy. The authors developed a model that can trigger seizures "on demand" using optogenetic stimulation of CA1 principal cells in mice rendered epileptic by an intra-hippocampal kainate (IHK) injection into CA3. The authors discuss their model in the context of the limitations of current animal models used in epilepsy drug development. In particular, their model addresses concerns regarding existing models where testing typically involves inducing acute seizures in healthy animals or waiting on infrequent, spontaneous seizures in epileptic animals.

Strengths:

A strength of this manuscript is that this approach may facilitate the evaluation of novel therapeutics since these evoked seizures, despite having some features that were significantly different from spontaneous seizures, are suggested to be sufficiently similar to spontaneous seizures which are more laborious to analyze. The data demonstrating the commonality of pharmacology and EEG features between evoked seizures and spontaneous seizures in epileptic mice, while also being different from evoked seizures in naïve mice, are convincing. The structural, functional, and behavioral differences between a seizure-naïve and epileptic mouse, which emerge due to the enduring changes occurring during epileptogenesis, are complex and important. Accordingly, this study highlights the importance of using mice that have underwent epileptogenesis as model organisms for testing novel therapeutics. Furthermore, this study positively impacts the wider epilepsy research community by investigating seizure semiology in these populations.

Weaknesses:

This study convincingly demonstrates that the feature space measurements for stimulus-evoked seizures in epileptic mice were significantly different from those in naïve mice; this result allows the authors to conclude that "seizures induced in chronically epileptic animals differed from those in naïve animals". However, the authors also conclude that "induced seizures resembled naturally occurring spontaneous seizures in epileptic animals" despite their own data demonstrating similar, albeit fewer, significant differences in feature space measurements. It is unclear if and what the threshold is whereby significant differences in these feature space measurements lead to the conclusion that the differences are meaningful, as in the comparison of epileptic and naïve mice, or not meaningful, as in the comparison of evoked and spontaneous seizures.

---

## [Referee Report · Reviewer #2 (Public review)]

The authors aimed to develop an animal model of temporal lobe epilepsy (TLE) that will generate "on-demand" seizures and an improved platform to advance our ability to find new anti-seizure drugs (ASDs) for drug-resistant epilepsy (DRE). Unlike some of the work in this field, the authors are studying actual seizures, and hopefully events that are similar to actual epileptic seizures. To develop an optimized screening tool, however, one also needs high-throughput systems with actual seizures as a quantitative, rigorous, and reproducible outcome measures. The authors aim to provide such a model; however, this approach may be over-stated here and seems unlikely to address the critical issue of drug resistance, which is their most important claim.

Strengths:

- The authors have generated an animal model of "on demand" seizures, which could be used to screen new ASDs and potentially other therapies. The authors and their model make a good-faith effort to emulate the epileptic condition and to use seizure susceptibility or probability as a quantitative output measure.

- The events considered to be seizures appear to be actual seizures, with some evidence that the seizures are different from seizures in the naïve brain. Their effort to determine how different ASDs raise seizure probability or threshold to an optogenetic stimulus to the CA1 area of the rodent hippocampus is focused on an important problem, as many if not most ASD screening uses surrogate measures that may not be as well linked to actual epileptic seizures.

- Another concern is their stimulation of dorsal hippocampus, while ventral hippocampus would seem more appropriate.

- Use of optogenetic techniques allows specific stimulation of the targeted CA1 pyramidal cells, and it appears that this approach is reproducible and reliable with quantitative rigor.

- The authors have taken on a critically important problem, and have made a good-faith effort to address many of the technical concerns raised in the reviews, but the underlying problem of DRE remains.

Weaknesses:

- Although the model has potential advantages, it also has disadvantages. As stated by the authors, the pre-test work-load to prepare the model may not be worth the apparent advantages. And most important, the paper frequently mentions DRE but does not directly address it, and yet drug resistance is the critical issue in this field.

- Although the paper shows examples of actual seizures, there remains some concern that some of the events might not be seizures - or a homogeneous population of seizures. More quantitative assessment of the electrical properties (e.g., duration) of the seizures and their probability is likely to be more useful than the proposed quantification in the future of the behavioral seizure stages, because the former could be both more objective and automated, while the behavioral analysis of the seizures will likely be more subjective and less reliable (and also fraught with subjectivity and analytical problems). Nonetheless, the authors point that the presence of "Racine 3 or above" behavioral seizures (in addition to their electrical data) is a good argument that many (if not all) of the "seizures" are actual epileptic seizures.

- Optogenetic stimulation of CA1 provides cell-specificity for the stimulation, but it is not clear that this method would actually be better than electrical stimulation of a kindled rodent with superimposed hippocampal injury. The reader is unfortunately left with the concern of whether this model would be easier and more efficacious than kindling.

- Although the authors have taken on a critically important problem, and have combined a variety of technologies, this approach may facilitate more rapid screening of ASDs against actual seizures (beneficial), but it does not really address the fundamentally critical yet difficult problem of DRE. A critical issue for DRE that is not well-addressed relates to adverse effects, which is often why many ASDs are not well tolerated by many patients (e.g., LEV). Thus, we are left with: how does this address anti-seizure DRE?

- The focus of this paper seems to be more on seizures more than on epilepsy. In the absence of seizure spontaneity, the work seems to primarily address the issues of seizure spread and duration. Although this is useful, it does not seem to be addressing the question of what trips the system to generate a seizure.

An appraisal of whether the authors achieved their aims, and whether the results support their conclusions:

- The authors seem to have developed a new and useful model; however, it is not clear how this will address that core problem of DRE, which was their stated aim.

- A discussion of the likely impact of the work on the field, and the utility of the methods and data to the community.

- As stated before in the original review, the potential impact would primarily be aimed at the ETSP or a drug-testing CRO; however, much more work will be required to convince the epilepsy community that this approach will actually identify new ASDs for DRE. The approach is potentially time-consuming with a steep and potentially difficult optimization curve, and thus may not be readily adaptable to the typical epilepsy-models neuroscience laboratory.

Any additional context you think would help readers interpret or understand the significance of the work:

- The problem of DRE is much more complicated than described by the authors here; however, the paper could end up being more useful than is currently apparent. Although this work could be seen as technically - and maybe conceptually - elegant and a technical tour de force, will it "deliver on the promise"? Is it better than kindling for DRE? In attempting to improve the discovery process, how will the model move us to another level? Will this model really be any better than others, such as kindling?

---

## [Referee Report · Reviewer #3 (Public review)]

This revised paper develops and characterizes a new approach for screening drugs for epilepsy. The idea is to increase the ability to study seizures in animals with epilepsy because most animal models have rare seizures. Thus, the authors use the existing intrahippocampal kainic acid (IHKA) mouse model, which can have very unpredictable seizures with long periods of time between seizures. This approach is of clear utility to researchers who may need to observe many seizure events per mouse during screening of antiseizure medications. A key strength is also that more utility can be derived from each individual mouse. The authors modified the IHKA model to inject KA into CA3 instead of CA1 in order to preserve the CA1 pyramidal cells that they will later stimulate. To express the excitatory opsin channelrhodopsin (ChR2) in area CA1, they use a virus that expresses ChR2 in cells that express the Thy-1 promoter. The authors demonstrate that CA3 delivery of KA can induce a very similar chronic epilepsy phenotype to the injection of KA in CA1 and show that optical excitation of CA1 can reliably induce seizures. The authors evaluate the impact of repeated stimulation on the reliability of seizure induction and show that seizures can be reliably induced by CA1 stimulation, at least for the short term (up to 16 days). These are strengths of the study.

However, there are several limitations: the seizures are evoked, not spontaneous. It is not clear how induced seizures can be used to investigate if antiseizure medication can reduce spontaneous seizures. Although seizure inducibility and severity can be assessed, the lack of spontaneous seizures is a limitation. To their credit, the authors show that electrophysiological signatures of induced vs spontaneous seizures are similar in many ways, but the authors also show several differences. Notably, the induced seizures are robustly inhibited by the antiseizure medication levetiracetam and variably but significantly inhibited by diazepam, similar to many mouse models with chronic recurrent seizure activity. One also wonders if using a mouse model with numerous seizures (such as the pilocarpine model) might be more efficient than using a modified IHKA protocol.

In this revised manuscript, the authors address some previous concerns related to definitions of seizures and events that are trains of spikes, sex as a biological variable, and present new images of ChR2 expression (but these images could be improved to see the cells more clearly). A few key concerns remain unaddressed, however. For example, it is still not clear that evoked seizures triggered by stimulating CA1 are similar to spontaneous seizures, regardless of the idea that CA1 plays a role in seizure disorders. It also remains unclear whether repeated activation of the hippocampal circuit will result in additional alterations to this circuit that affect the seizure phenotype over prolonged intervals (after 16 days). Furthermore, the use of SVM with the number of seizures being used as replicates (instead of number of mice) is inappropriate. Another theoretical concern is whether the authors are correct in suggesting that one will be able to re-use the mice for screening multiple drugs in a row.

Strengths:

- The authors show that the IHKA model of chronic epilepsy can be modified to preserve CA1 pyramidal cells, allowing optogenetic stimulation of CA1 to trigger seizures.

- The authors show that repeated optogenetic stimulation of CA1 in untreated mice can promote kindling and induce seizures, indeed generating two mouse models in total.

- Many electrophysiological signatures are similar between the induced and spontaneous seizures, and induced seizures reliably respond to treatment with antiseizure medications.

- Given that more seizures can be observed per mouse using on-demand optogenetics, this model enhances the utility of each individual mouse.

- Mice of each sex were used.

Weaknesses:

- Evaluation of seizure similarity using the SVM modeling and clustering is not sufficiently justified when using number of seizures as the statistical replicate (vs mice).

- Related to the first concern, the utility of increasing number of seizures for enhancing statistical power is limited because standard practice is for sample size to be numbers of mice.

- The term "seizure burden" usually refers to the number of spontaneous seizures per day, not the severity of the seizures themselves. Because the authors are evoking the seizures being studied, this study design precludes assessment of seizure burden.

- It seems likely that repeatedly inducing seizures will have a long-term effect, especially in light of the downward slope at day 13-16 for induced seizures seen in Figure 4C. A duration of evaluation that is longer than 16 days is warranted.

- Human epilepsy is extensively heterogeneous in both etiology and individual phenotype, and it may be hard to generalize the approach.

---

## [Author Response]

The following is the authors’ response to the original reviews

**Reviewer 1 (Public review):**
Weaknesses:While the data generally supports the authors' conclusions, a weakness of this manuscript lies in their analytical approach where EEG feature-space comparisons used the number of spontaneous or evoked seizures as their replicates as opposed to the number of IHK mice; these large data sets tend to identify relatively small effects of uncertain biological significance as being highly statistically significant. Furthermore, the clinical relevance of similarly small differences in EEG feature space measurements between seizure-naïve and epileptic mice is also uncertain.

In this work, we used linear mixed effect model to address two levels of variability –between animals and within animals. The interactive linear mixed effect model shows that most (~90%) of the variability in our data comes from within animals (Residual), the random effect that the model accounts for, rather than between animals. Since variability between animals are low, the model identifies common changes in seizure propagation across animals, while accounting for the variability in seizures within each animal. Therefore, the results we find are of changes that happen across animals, not of individual seizures. We made text edits to clarify the use of the linear mixed effect model. (page6, second paragraph and page 11, first paragraph)

Finally, the multiple surgeries and long timetable to generate these mice may limit the value compared to existing models in drug-testing paradigms.

Thank you for the suggestion. We added a discussion in the ‘Comparison to other seizure models…’ section on pages 15 and 16. In an existing model investigating spontaneous tonic-clonic seizures (such as the intra-amygdala kainate injection model), the time investment is back-loaded, requiring two to three weeks per condition while counting spontaneous seizures, which may occur only once a day. In contrast, our model requires a front-loaded time investment. Once the animals are set up, we can test multiple drugs within a few weeks, providing significant time savings. Additionally, we did not pre-screen animals in our study. Existing models often pre-select mice with high rates of spontaneous seizures, whereas in our model, seizures can be induced even in animals with few spontaneous seizures. We believe that bypassing the need for pre-screening also is a key advantage of our induced seizure model.

**Reviewer 1 (Recommendations for the authors):**
(1) Address why the EEG data comparisons were performed between seizures and not between animals (as explicitly described in the public review). Further, a discussion of the biological significance (or lack thereof) of the effect size differences observed is warranted. This is especially concerning when the authors make the claim that spontaneous and induced seizures are essentially the same while their analysis shows all evaluated feature space parameters were significantly difference in the initial 1/3 of the EEG waveforms.

We made text edits to clarify the use of the linear mixed effects model (page 6, second paragraph, and page 11, first paragraph)

(2) The authors place great emphasis on the use of clinically/etiologically relevant epilepsy models in drug discovery research. There is discussion criticizing the time points required to enact kindling and the artificial nature of acute seizure induction methods. However, the combination IHK-opto seizure induction model also requires a lengthy timeline. A more tempered discussion of this novel model's strengths may benefit readers.

Thank you for the suggestion. We added a discussion in the ‘Comparison to other seizure models…’ section on pages 15 and 16.

(3) The authors should further emphasize the benefit of having an inducible seizure model of focal epilepsy since other mouse models (e.g., genetic or TBI models) may have superior etiological relevance (construct and face validity) but may not be amenable to their optogenetic stimulation approach.

Thank you for the suggestion. We revised the manuscript to better emphasize the potential significance of our approach. We added a discussion in the 'Application of Models...' section on page 15, second paragraph. The on-demand seizure model can be applied to address biologically and clinically relevant questions beyond its utility in drug screening. For example, crossing the Thy1-ChR2 mouse line with genetic epilepsy models, such as Scn1a mutants, could reveal how optogenetic stimulation differentially induces seizures in mutant versus non-mutant mice, providing insights into seizure generation and propagation in Dravet syndrome. Due to the cellular specificity of optogenetics, we also envision this approach being used to study circuit-specific mechanisms of seizure generation and propagation.

(4) Suggestion: Provide immunolabeled imagery demonstrating ChR2 presence in Thy1 cells.

Thank you for the suggestion. We added a fluorescence image showing ChR2 expression in Fig. 2A

(5) It might be prudent to mention any potential effects of laser heat on hippocampal cell damage, although the 10 Hz, ~10 mW, and 6 s stim is unlikely to cause any substantial burns. Without knowing the diameter and material of the optic fiber, this is left up to some interpretation.

Thank you for the comments. In the Methods section, we listed the optical fiber diameter as 400 microns (page 17, EEG and Fiber Implantation section). Using 5–18 mW laser power with a relatively large fiber diameter of 400 microns, the power density falls within the range of commonly employed channelrhodopsin activation conditions in vivo. That said, we would like to investigate potential heat effects or cell damage in a follow-up study.

(6) There are instances in the manuscript where the authors describe experimental and analytical parameters vaguely (e.g. "Seizures were induced several times a day", "stimulation was performed every 1 - 3 hours over many days"). These descriptions can and should be more precise.

Thank you for the comments. To enhance clarity, we added the stimulation protocol in a flowchart format in Fig. S2A, describing how we determined the threshold and proceeded to the drug test. Following this protocol, there was variability in the number of stimulations per day.

(7) In the second to last paragraph of the discussion, the authors state "However, HPDs are not generalizable across species - they are specific to the mouse model (55)." This statement is inaccurate. The paper cited comes from Dr. Corrine Roucard's lab at Synapcell. In fact, Dr. Rouchard argues the opposite (See Neurochem Res (2017) 42:1919-1925).

Thank you for pointing out the mistake. On page 16, in the first paragraph, reference 55 (now 58 in the revised version) was intended to refer to 'quickly produce dose-response curves with high confidence.' In the revision, we cited another paper reporting that hippocampal spikes were not reproduced in the rat IHK model. R. Klee, C. Brandt, K. Töllner, W. Löscher, Various modifications of the intrahippocampal kainate model of mesial temporal lobe epilepsy in rats fail to resolve the marked rat-to-mouse differences in type and frequency of spontaneous seizures in this model. Epilepsy Behav. 68, 129–140 (2017).

(8) In the discussion, Levetiracetam is highlighted as an ASM that would not be detected in acute induced seizure models; the authors point out its lack of effect in MES and PTZ. However, LEV is effective in the 6Hz test (also an acute-induced seizure model). This should be stated.

Thank you for the comments. We highlighted the discussion on LEV in the 'Application of Model to Testing Multiple Classes of ASMs...' section on page 14.

(9) The results text indicates that 9 epileptic mice were used to test LEV and DZP. However, the individual data points illustrated in Figure 5B show N=8 mice. Please correct.

Thank you for the comments. A total of nine epileptic mice were used to assess two drugs, with the animals being re-used as indicated in the schematic. A total of eight assessments were conducted for DZP with six mice and eight assessments for LEV with five mice. Each assessment included hourly ChR2 activations without an ASM and hourly ChR2 activations after ASM injection.

(10) Figure 4D: Naïve mice are labeled as solid blue circles in the legend while the data points are solid blue triangles. Please correct.

Thank you. We corrected the marker in Fig.4D.

**Reviewer 2 (Public Review):**
Weaknesses:(1) Although the figures provide excellent examples of individual electrographic seizures and compare induced seizures in epileptic and naïve animals, it is unclear which criteria were used to identify an actual seizure induced by the optogenetic stimulus, versus a hippocampal paroxysmal discharge (HPD), an "afterdischarge", an "electrophysiological epileptiform event" (EEE, Ref #36, D'Ambrosio et al., 2010 Epilepsy Currents), or a so-called "spike-wave-discharge" (SWD). Were HPDs or these other non-seizure events ever induced using stimulation in animals with IH-KA? A critical issue is that these other electrical events are not actual seizures, and it is unclear whether they were included in the column showing data on "electrographic afterdischarges" in Figure 5 for the studies on ASDs. This seems to be a problem in other areas of the paper, also.

Thank you for pointing out the unclear definition of the seizures analyzed. We added sentences at the beginning of the Results section (page 3) to clarify the terminology we used. We analyzed animal behavior during evoked events, and a high percentage of induced electrographic events were accompanied by behavioral seizures with a Racine scale of three or above. We added Supplemental Figure S9, which shows behavioral seizure severity scores observed before and during ASM testing. We hope these changes address the reviewer’s concern and improve the clarity of the manuscript.

(2) The differences between the optogenetically evoked seizures in IH-KA vs naïve mice are interpreted to be due to the "epileptogenesis" that had occurred, but the lesion from the KA-induced injury would be expected to cause differences in the electrically and behaviorally recorded seizures - even if epileptogenesis had not occurred. This is not adequately addressed.

Thank you for the comments. IHK-injected mice had spontaneous tonic-clonic seizures before the start of optical stimulation, as shown in Figure S1.

(3) The authors offer little mention of other research using animal models of TLE to screen ASDs, of which there are many published studies - many of them with other strengths and/or weaknesses. For example, although Grabenstatter and Dudek (2019, Epilepsia) used a version of the systemic KA model to obtain dose-response data on the effects of carbamazepine on spontaneous seizures, that work required use of KA-treated rats selected to have very high rates of spontaneous seizures, which requires careful and tedious selection of animals. The ETSP has published studies with an intra-amygdala kainic acid (IA-KA) model (West et al., 2022, Exp Neurol), where the authors claim that they can use spontaneous seizures to identify ASDs for DRE; however, their lack of a drug effect of carbamazepine may have been a false negative secondary to low seizure rates. The approach described in this paper may help with confounds caused by low or variable seizure rates. These types of issues should be discussed, along with others.

We appreciate the reviewer’s insights. We added a discussion comparing our model with other existing models in the Discussion section (pages 15 and 16, 'Comparison to Other Seizure Models Used in Pharmacologic Screening' section). In an existing model investigating spontaneous tonic-clonic seizures (such as the intra-amygdala kainate injection model), the time investment is back-loaded, requiring two to three weeks per condition while counting spontaneous seizures, which may occur only once a day. In contrast, our model requires a front-loaded time investment. Once the animals are set up, we can test multiple drugs within a few weeks, providing significant time savings. Additionally, we did not pre-screen animals in our study. Existing models often pre-select mice with high rates of spontaneous seizures, whereas in our model, seizures can be induced even in animals with few spontaneous seizures. We believe that bypassing the need for pre-screening is a key advantage of our induced seizure model.

(4) The outcome measure for testing LEV and DZP on seizures was essentially the fraction of unsuccessful or successful activations of seizures, where high ASD efficacy is based on showing that the optogenetic stimulation causes fewer seizures when the drug is present. The final outcome measure is thus a percentage, which would still lead to a large number of tests to be assured of adequate statistical power. Thus, there is a concern about whether this proposed approach will have high enough resolution to be more useful than conventional screening methods so that one can obtain actual dose-response data on ASDs.

Thank you for the comments. In this revision, we added Supplemental Figure S9, showing the severity of behavioral seizures observed before and during ASM testing for each animal. We observed a reduction in behavioral seizure severity for each subject. We would like to explore using behavioral severity as an outcome measure in a follow-up study.

(5) The authors state that this approach should be used to test for and discover new ASDs for DRE, and also used for various open/closed loop protocols with deep-brain stimulation; however, the paper does not actually discuss rigorously or critically the background literature on other published studies in these areas or how this approach will improve future research for a broader audience than the ETSP and CROs. Thus, it is not clear whether the utility will apply more widely and how extensive a readership will be attracted to this work.

We appreciate the reviewer’s insights. We revised the manuscript to better emphasize the potential significance of our approach (page 15, second paragraph). The on-demand seizure model can be applied to address biologically and clinically relevant questions beyond its utility in drug screening. For example, crossing the Thy1-ChR2 mouse line with genetic epilepsy models, such as Scn1a mutants, could reveal how optogenetic stimulation differentially induces seizures in mutant versus non-mutant mice, providing insights into seizure generation and propagation in Dravet syndrome. Due to the cellular specificity of optogenetics, we also envision this approach being used to study circuit-specific mechanisms of seizure generation and propagation. Regarding drug-resistant epilepsy (DRE) and anti-seizure drug (ASD) screening, we agree with the reviewer that probing new classes of ASDs for DRE represents a critical goal. However, we believe that a full exploration of additional ASD classes and/or modeling DRE lies outside the scope of this manuscript, and we would like to explore it in a follow-up study.

**Reviewer 2 (Recommendations for the authors):**
(1) The authors should explain why 10 Hz was chosen as the stimulation frequency.

Thank you for the comment. A frequency of 10 Hz was determined based on previous work using anesthetized animals prepared in an acute in vivo setting. To simplify the paper and avoid confusion, we did not include a discussion on how we determined the frequency. Instead, we added a detailed description of how we optimized the power in a flowchart format in Supplemental Figure S2. We hope this improves reproducibility.

(2) After micro-injection of KA, morphological changes were observed in the hippocampus, but no comparison of Chr2 expression was made in naïve animals vs KA-injected animals. Presumably, the Thy1-Chr2 mouse expresses GFP in cells that express Chr2. Thus, it may be useful to show the expression of Chr2 in animals with hippocampal sclerosis. This may explain the lack of dramatic difference between stimulation parameters in naïve vs epileptic animals, as shown in supplemental Figure S2.

Thank you for the suggestion. We added a fluorescence image of ChR2 expression in CA1, ipsilateral to the KA-injected site, in Fig. 2A.

(3) The authors state that "During epileptogenesis, neural networks in the brain undergo various changes ranging from modification of membrane receptors to the formation of new synapses" and that these changes are critical for successful "on-demand" seizure induction. However, it is not clear or well-discussed whether changes in neuronal cell densities that occur during sclerosis are important for "on-demand" seizure induction as well. Also, the authors showed that naïve animals exhibit a kindling-like effect, but it was unclear whether a similar effect was present in epileptic animals (i.e. do stimulation thresholds to seizure induction change as the animal gets more induction stimulations)? If present, would the secondary kindling affect drug-testing studies (e.g., would the drug effect be different on induced seizure #2 vs induced seizure #20)?

Thank you for the suggestion. Since this is an important aspect of the model, we would like to address the kindling effect, the secondary kindling effect, and histopathology in a longer-term setting (several weeks) in a follow-up study.

(4) The authors show that in their model, LEV and DZP were both efficacious. The authors do not seem to mention that, over 25 years ago, LEV was originally missed in the standard ETSP screens; and, it was only discovered outside of the ETSP with the kindling model. The kindling model is now used to screen ASDs. The authors should consider adding this point to the Discussion. It remains unclear, however, if the author's screening strategy shows advantages over kindling and other such approaches in the field.

Thank you for the suggestion. We added a discussion on LEV in the 'Application of Model to Testing Multiple Classes of ASMs...' section on page 14.

(5) P8 paragraph 2. The authors state values for naïve animals, but they should also provide values for epileptic animals since they state that the groups were not significantly different (p>0.05). It would be useful to show values for both and state the actual p-value from the test. This issue of stating mean/median values with SD and sample size should be addressed for all data throughout the paper. Additionally, Figure S2 should be added to the manuscript and discussed, as it has data that may be valuable for the reproducibility of the paper.

Thank you for the suggestion. Figure S2 shows the threshold power required to induce electrographic activity for n = 10 epileptic animals (9.14 ± 4.75 mW) and n = 6 naïve animals (6.17 ± 1.58 mW) (Wilcoxon rank-sum test, p = 0.137). The threshold duration was comparable between the same epileptic animals (6.30 ± 1.64 s) and naïve animals (5.67 ± 1.03 s) (Wilcoxon rank-sum test, p = 0.7133).

(6) In addition to the other stated references on synaptic reorganization in the CA1 area, the authors should mention similar studies from Esclapez et al. (1999, J Comp Neurol).

Thank you. We have included the reference in the revision.

(7) All of the raw EEG data on the seizures should be accessible to the readers.

Thank you for the suggestion. We will consider depositing EEG data in a publicly accessible site.

**Reviewer 3 (Public review):**
Weaknesses:(1) Evaluation of seizure similarity using the SVM modeling and clustering is not sufficiently explained to show if there are meaningful differences between induced and spontaneous seizures. SVM modeling did not include analysis to assess the overfitting of each classifier since mice were modeled individually for classification.”

Thank you for the comment. We made text edits to clarify the purpose of the SVM analysis. It was not intended to identify meaningful differences between induced and spontaneous seizures. Rather, it was used to classify EEG epochs as 'seizures' based on spontaneous seizures as the training set, demonstrating the gross similarity between induced and spontaneous seizures.

(2) The difference between seizures and epileptiform discharges or trains of spikes (which are not seizures) is not made clear.

Thank you for pointing out the unclear definition of the seizures analyzed. We added sentences at the beginning of the Results section (page 3) to clarify the terminology we used. We analyzed animal behavior during evoked events, and a high percentage of induced electrographic events were accompanied by behavioral seizures with a Racine scale of three or above. We added Supplemental Figure S9 to show the types of seizures observed before and during ASM testing. We hope these changes address the reviewer’s concern and improve the clarity of the manuscript.

(3) The utility of increasing the number of seizures for enhancing statistical power is limited unless the sample size under evaluation is the number of seizures. However, the standard practice is for the sample size to be the number of mice.

In this work, we used a linear mixed-effects model to address two levels of variability—between animals and within animals. The interactive linear mixed-effects model shows that most (~90%) of the variability in our data comes from within animals (residual), the random effect that the model accounts for, rather than between animals. Since variability between animals is low, the model identifies common changes in seizure propagation across animals while accounting for the variability in seizures within each animal. Therefore, the results we find reflect changes that occur across animals, not individual seizures. We made text edits to clarify the use of the linear mixed-effects model.

(4) Seizure burden is not easily tested.

Thank you for the comment. We added Supplemental Figure S9 to summarize the severity of behavioral seizures before and during ASM testing. This addresses the reviewer’s comment on seizure burden. In a follow-up study, we would like to explore this type of outcome measure for drug screening.

**Reviewer 3 (Recommendations for the authors):**
(1) Provide a stronger rationale to use area CA1. For example, the authors mention that CA1 is active during seizure activity, but can seizures originate from CA1? That would make the approach logical and also explain why induced and spontaneous seizures are similar.

Thank you for the comment. We discussed it in the Discussion section (page 14, first and second paragraphs).

(2) Explain the use of SVM classifiers so it is more convincing that induced and spontaneous seizures are similar. Or, if they are not similar, explain that this is a limitation.

We made text edits to clarify the purpose of the SVM analysis. It was not intended to identify meaningful differences between induced and spontaneous seizures. Rather, it was used to classify EEG epochs as 'seizures' based on spontaneous seizures as the training set, demonstrating the gross similarity between induced and spontaneous seizures.

(3)If feasible, extend the duration over which seizure induction reliability is assessed so that the long-term utility of the model can be demonstrated.

Thank you for the suggestion. We would like to assess long-term utility in a follow-up study.

(4) The GitHub link is not yet active. The authors will be required to supply their relevant code for peer evaluation as well as publication.

Thank you. The GitHub repository is now active.

(5) State and assess the impacts of sex as a biological variable.

Thank you for pointing this out. Both female and male animals were included in this study: Epileptic cohort: 7 males, 3 females; Naïve cohort: 3 males, 4 females.